# Multi-modal brain encoding models for multi-modal stimuli

## Abstract

Despite participants engaging in single modality stimuli, such as watching images or silent videos, recent work has demonstrated that multi-modal Transformer models can predict visual brain activity impressively well, even with incongruent modality representations. This raises the question of how accurately these multi-modal models can predict brain activity when participants are engaged in multi-modal stimuli. As these models grow increasingly popular, their use in studying neural activity provides insights into how our brains respond to such multi-modal naturalistic stimuli, i.e., where it separates and integrates information from different sensory modalities. We investigate this question by using multiple unimodal and two types of multi-modal models—cross-modal and jointly pretrained—to determine which type of models is more relevant to fMRI brain activity when participants were engaged in watching movies (videos with audio). We observe that both types of multi-modal models show improved alignment in several language and visual regions. This study also helps in identifying which brain regions process unimodal versus multi-modal information. We further investigate the impact of removal of unimodal features from multi-modal representations and find that there is additional information beyond the unimodal embeddings that is processed in the visual and language regions. Based on this investigation, we find that while for cross-modal models, their brain alignment is partially attributed to the video modality; for jointly pretrained models, it is partially attributed to both the video and audio modalities. The inability of individual modalities in explaining the brain alignment effectiveness of multi-modal models suggests that multi-modal models capture additional information processed by all brain regions. This serves as a strong motivation for the neuro-science community to investigate the interpretability of these models for deepening our understanding of multi-modal information processing in brain.

## 1  Introduction

The study of brain encoding aims at predicting the neural brain activity recordings from an input stimulus representation. Recent brain encoding studies use neural models as a powerful approach to better understand the information processing in the brain in response to naturalistic stimuli (Oota et al., 2023a). Current encoding models are trained and tested on brain responses captured from participants who are engaged in a *single stimulus modality*, using stimulus representations extracted from AI systems that are pretrained on single modality, such as language (Wehbe et al., 2014; Jain & Huth, 2018; Toneva & Wehbe, 2019; Caucheteux & King, 2020; Schrimpf et al., 2021; Toneva et al., 2022; Aw & Toneva, 2023), vision (Yamins et al., 2014; Eickenberg et al., 2017; Schrimpf et al., 2018; Wang et al., 2019) or speech (Millet et al., 2022; Vaidya et al., 2022; Tuckute et al., 2023). In this paper, we build encoding models where participants are engaged with *multi-modal stimuli* (e.g., watching movies that also include audio). We explore multi-modal stimulus representations extracted

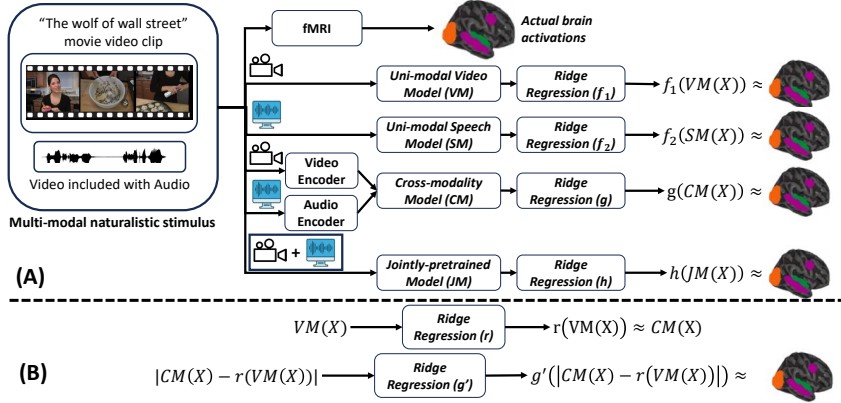

Figure 1: (A) Overview of our proposed Multi-modal Brain Encoding Pipeline. (B) Residual Analysis.

using Transformer (Vaswani et al., 2017) based multi-modal models. Our analysis focuses on their alignment with both uni- and multi-modal brain regions.

There is a growing evidence that the human brain's ability for multi-modal processing is underpinned by synchronized cortical representations of identical concepts across various sensory modalities (Gauthier et al., 2003; Bracci & Op de Beeck, 2023). Reflecting similar principles, the recent advances in AI systems have led to the development of multi-modal models (like CLIP (Radford et al., 2021), ImageBind (Girdhar et al., 2023), and TVLT (Tang et al., 2022)) using massive interleaved image-text data, speech-text data or video-audio-text data to represent multi-modal input. This recent progress in AI has stimulated advancements in brain encoding models (Doerig et al., 2022; Oota et al., 2022; Popham et al., 2021; Wang et al., 2022; Tang et al., 2024; Nakagi et al., 2024) that learn effectively from multiple input modalities, despite participants being engaged with single stimulus modality during experiments, e.g., watching natural scene images, or silent movie clips. However, these studies have experimented with subjects engaged with single-modality stimulus, leaving the full potential of these models in true multi-modal scenarios still unclear.

Using brain recordings of participants watching several popular movies included with audio (St-Laurent et al., 2023), we investigate several research questions. First, we investigate the effectiveness of multi-modal stimulus representations obtained using multi-modal models versus unimodal models for brain encoding. Multi-modal models are of two broad types: (i) cross-modal pretrained models, where first individual modality encoders are trained and then cross-modal alignment is performed, and (ii) jointly pretrained models, which involve combining data from multiple modalities and training a single joint encoder. Hence, we also investigate which of the two types (cross-modal versus joint) are better for encoding. In this work, we focus on one cross-modal (ImageBind), one jointly pretrained (TVLT), three video and two speech models. Additionally, we explore which modality representations are more brain relevant, and identify which brain regions process uni- and multi-modal information. Overall, this research utilizes various modality representations to develop encoding models based on fMRI responses within a multi-modal model framework (see Fig. 1 for workflow).

Using our multi-modal brain encoding approach, we examine several insights. First, we use previous neuroscience findings that have identified brain regions involved in visual, language and auditory processing, and investigate how well our model aligns with these regions when both the model and a human participant watch the same multi-modal video stimuli. Second, we expect that multi-modal models which can learn cross-modal and joint embeddings across modalities in a brain-relevant way would significantly align with these regions. However, alignment with these brain regions doesn't necessarily mean that the model is effectively learning from multiple modalities, as unimodal models for vision or language or audio have also been shown to significantly align with these brain regions (Wehbe et al., 2014; Toneva et al., 2022; Schrimpf et al., 2021; Millet et al., 2022; Vaidya et al., 2022). To check the second aspect, we investigate this question via a direct approach, closely related to previous studies (Toneva et al., 2022; Oota et al., 2023b,c). For each modality, we analyze how the alignment between brain recordings and multi-modal model representations is affected by the elimination of information related to that particular modality from the model representation.

Our analysis of multi-modal brain alignment leads to several key conclusions: (1) Both cross-modal and jointly pretrained models demonstrate significantly improved brain alignment with language

regions (AG, PCC, PTL, and IFG) and visual regions (EVC and MT) when analyzed against unimodal video data. In contrast, compared to unimodal speech-based models, all multi-modal embeddings show significantly better brain alignment, except in the OV (object visual processing) region. This highlights the ability of multi-modal models to capture additional information—either through knowledge transfer or integration between modalities—which is crucial for multi-modal brain alignment. (2) Using our residual approach, we find that the improved brain alignment in cross-modal models can be partially attributed to the removal of video features alone, rather than auditory features. On the other hand, the improved brain alignment in jointly pretrained models can be partially attributed to the removal of both video and auditory features.

Overall, we make the following contributions in this paper. (1) To the best of our knowledge, this study is the first to leverage both cross-modal and jointly pretrained multi-modal models to perform brain alignment while subjects are engaged with multi-modal naturalistic stimuli. (2) We evaluate the performance of several unimodal Transformer models (three video and two audio) and measure their brain alignment. (3) Additionally, we remove unimodal features from multi-modal representations to explore the impact on brain alignment before and after their removal. We will release code upon publication of this paper.

## 2 Related Work

**Multi-modal models.** Pretrained Transformer-based models have been found to be very effective in various tasks related to language (Devlin et al., 2019; Radford et al., 2019), speech (Baevski et al., 2020), and images (Dosovitskiy et al., 2020). To learn associations between pairs of modalities, Transformer models have been pretrained on multiple modalities, showing excellent results in multi-modal tasks like visual question answering and visual common-sense reasoning. These multi-modal models are pretrained in two different ways: (i) cross-modal models that integrate information from multiple modalities and learn a joint encoder, such as VisualBERT (Li et al., 2019) and ImageBind (Girdhar et al., 2023), and (ii) jointly pretrained models like LXMERT (Tan & Bansal, 2019), CLIP (Radford et al., 2021), ViLBERT (Lu et al., 2019), and TVLT (Tang et al., 2022) which fuse individual modality encoders at different stages, transferring knowledge from one modality to another. In this work, we investigate how the representations extracted from *cross-modal and jointly-pretrained Transformer models* align with human brain recordings when participants engage with multi-modal stimuli.

**Brain Encoding using Multi-modal Models.** Since human brain perceives the environment using information from multiple modalities (Gauthier et al., 2003), examining the alignment between language and visual representations in the brain by training encoding models on fMRI responses, while extracting joint representations from multi-modal models, can offer insights into the relationship between the two modalities. For instance, it has been shown that multi-modal models like CLIP (Radford et al., 2021) better predict neural responses in the high-level visual cortex as compared to previous vision-only models (Doerig et al., 2022; Wang et al., 2022). Additionally, Tang et al. (2024) demonstrate the use of multi-modal models in a cross-modal experiment to assess how well the language encoding models can predict movie-fMRI responses and how well the vision encoding models can predict narrative story-fMRI. Nakagi et al. (2024) analyzed fMRI related to video content viewing and found distinct brain regions associated with different semantic levels, highlighting the significance of modeling various levels of semantic content simultaneously. However, these studies have experimented with subjects engaged with single-modality stimulus, leaving the full potential of these models in true multi-modal scenarios still unclear. Recently, Dong & Toneva (2023) interpreted the effectiveness of pretrained versus finetuned multi-modal video transformer using video+text stimuli-based brain activity. However, they did not perform any cross-modal vs jointly-pretrained model analysis or analysis of multi-modal versus unimodal models, leaving it unclear which type of multi-modal models perform best for brain activity prediction. Further, unlike them, we study video+audio stimuli, and perform comprehensive residual analysis.

## 3 Dataset Curation

**Brain Imaging Dataset.** We experiment with a multi-modal naturalistic fMRI dataset, Movie10 (St-Laurent et al., 2023) obtained from the Courtois NeuroMod databank. This dataset was collected while six human subjects passively watched four different movies: *The Bourne supremacy (∼100*

*mins), The wolf of wall street (∼170 mins), Hidden figures (∼120 mins)* and *Life (∼50 mins)*. Among these, *Hidden figures* and *Life* are repeated twice, with the repeats used for testing and the remaining movies for training. In this work, we use *Life* movie for testing where we average the two repetitions to reduce noise in brain data. This dataset is one of the largest publicly available multi-modal fMRI dataset in terms of number of samples per participant. It includes 4024 TRs (Time Repetitions) for *The Bourne supremacy*, 6898 TRs for *The wolf of wall street* used in train and 2028 TRs for *Life* in test. The fMRI data is collected every 1.49 seconds (= 1TR).

The dataset is already preprocessed and projected onto the surface space ("fsaverage6"). We use the multi-modal parcellation of the human cerebral cortex based on the Glasser Atlas (which consists of 180 regions of interest in each hemisphere) to report the ROI (region of interest) analysis for the brain maps (Glasser et al., 2016). This includes four visual processing regions (early visual (EV), object-related areas (LO), face-related areas (OFA) and scene-related areas (PPA)), one early auditory area (AC), and eight language-relevant regions, encompassing broader language regions: angular gyrus (AG), anterior temporal lobe (ATL), posterior temporal lobe (PTL), inferior frontal gyrus (IFG), inferior frontal gyrus orbital (IFGOrb), middle frontal gyrus (MFG), posterior cingulate cortex (PCC) and dorsal medium prefrontal cortex (dmPFC), based on the Fedorenko lab's language parcels (Milton et al., 2021; Desai et al., 2022). We list the detailed sub-ROIs of these ROIs in Appendix B.

**Estimating dataset cross-subject prediction accuracy.** To account for the intrinsic noise in biological measurements, we adapt Schrimpf et al. (2021)'s method to estimate the cross-subject prediction accuracy for a model's performance for the Movie10 fMRI datasets. By subsampling fMRI datasets from 6 participants, we generate all possible combinations of $s$ participants ($s \in [2,6]$) for watching movies, and use a voxel-wise encoding model (see Sec. 5) to predict one participant's response from others. Note that the estimated cross-subject prediction accuracy is based on the assumption of a perfect model, which might differ from real-world scenarios, yet offers valuable insights into model's performance. We estimate cross-subject prediction accuracy in three settings: (i) training with *The Bourne supremacy* and testing with *Life* data, (ii) training with *The wolf of wall street* and testing with *Life* data, and (iii) training with both *The Bourne supremacy* and *The wolf of wall street* and testing with Life data. We present the average cross-subject prediction accuracy across voxels for the *Movie10 fMRI* dataset and across the three settings in Appendix A.

## 4 Methodology

### 4.1 Multi-modal models

To analyse how human brain process information while engaged in multi-modal stimuli, we use recent popular deep learning models to explore multiple modalities information and build the encoding models in two different ways: "cross-modality pretraining" and "joint pretraining".

**Cross-modality Pretrained Multi-modal Models.** Cross-modality representations involve transferring information or learning from one modality to another. For example, in a cross-modal learning scenario, text descriptions can be used to improve the accuracy of image/video recognition tasks. This approach is often used in scenarios where one modality might have limited data or less direct relevance but can be informed by another modality.

Recently, a cross-modal model called ImageBind (IB) (Girdhar et al., 2023) has shown immense promise in binding data from six modalities at once, without the need for explicit supervision. ImageBind model uses separate encoders for each individual modality and learns a single shared representation space by leveraging multiple types of image-paired data. ImageBind consists of 12 layers and outputs a 1024 dimensional representation for each modality.

**Jointly Pretrained Multi-modal Models.** Jointly pretrained multi-modal model representations, on the other hand, involve combining data from multiple modalities to build a more comprehensive joint understanding to improve decision-making processes. The system processes these diverse inputs concurrently to make more informed and robust decisions.

TVLT (Zellers et al., 2022) is an end-to-end Text-less Vision-Language multi-modal Transformer model for learning joint representations of video and speech from YouTube videos. This joint encoder model consists of a 12-layer encoder (hidden size 768) and uses masked autoencoding objective for both videos and speech. Given the video-speech pairs, the TVLT model provides 768 dimensional representations for each modality across 12 layers.

**Extraction of multi-modal features.** To extract video and audio embedding representations from multi-modal models for the brain encoding task, we input video and audio pairs at each TR and obtain the aligned embeddings for the two modalities. Here, we first segment the input video and audio into clips corresponding to 1.49 seconds, which matches the fMRI image rate. For both the models, ImageBind and TVLT, we use the pretrained Transformer weights. ImageBind generates an embedding for each modality (IB video and IB audio) in an aligned space. We concatenate these embeddings to create what we refer to as IB concat embeddings. On the other hand, TVLT provides a joint embedding across all modalities at each layer. Only for the last layer, TVLT provides an embedding for each modality.

## 4.2 Unimodal Models

To investigate the effectiveness of multi-modal representations in comparison to representations for individual modalities, we use the following methods to obtain embeddings for individual modalities.

**Video-based models.** To extract representations of the video stimulus, we use three popular pretrained Transformer video-based models from Huggingface (Wolf et al., 2020): (1) Vision Transformer Base (ViT-B) (Dosovitskiy et al., 2020), (2) Video Masked Autoencoders (VideoMAE) (Tong et al., 2022) and (3) Video Vision Transformer (ViViT) (Arnab et al., 2021). Details of each model are reported in Table 1 in Appendix.

**Speech-based models.** Similar to video-based models, we use two popular pretrained Transformer speech-based models from Huggingface: (1) Wav2Vec2.0 (Baevski et al., 2020) and (2) AST (Baade et al., 2022). Details of each model are reported in Table 1 in Appendix.

**Extraction of video features.** ViT-B (Dosovitskiy et al., 2020), the underlying video encoder model for ImageBind is used for extracting representations for all frames in each TR for every video. To extract embedding at each TR, we average all frame embeddings and obtain the corresponding video representation. For VideoMAE and ViViT, we directly obtain the video embeddings for each TR. All 3 models provide 768 dimensional representations and all of them are 12-layer Transformer encoders.

**Extraction of speech features.** To explore whether speech models incorporate linguistic information, we extract representations beyond 1.49 secs, i.e., we considered context window of 16 secs with stride of 100 msecs and considered the last token as the representative for each context window. The pretrained speech-based models output token representations at different layers. Both Wav2Vec2.0 and AST models provide 768 dimensional representations and all of them are 12-layer Transformer encoders. Finally, we align these representations with the fMRI data acquisition rate by downsampling the stimulus features with a 3-lobed Lanczos filter, thus producing chunk-embeddings for each TR.

## 5 Experimental Setup

**Encoding Model.** We train bootstrap ridge regression based voxel-wise encoding models (Deniz et al., 2019) to predict the fMRI brain activity associated with the stimulus representations obtained from the individual modalities (speech and video) and multi-modal embeddings from cross-modal and jointly pretrained multi-modal models. For each subject, we account for the delay in the hemodynamic response by modeling hemodynamic response function using a finite response filter (FIR) per voxel with 5 temporal delays (TRs) corresponding to $\sim$7.5 seconds (Huth et al., 2022). Formally, at each time step $t$, we encode the stimuli as $X_t \in \mathbb{R}^D$ and brain region voxels $Y_t \in \mathbb{R}^V$, where $D$ denotes the dimension of the concatenation of delayed 5 TRs, and $V$ denotes the number of voxels. Overall, with $N$ such TRs, we obtain $N$ training examples.

**Train-test Setup.** We build encoding models in three settings: (i) We used all data samples from 10 training sessions of the *The Bourne supremacy* movie for training and tested generalization on samples from the test sessions (5 sessions) of the *Life* movie. (ii) We used data from 17 training sessions of the *The wolf of wall street* movie for training, with the *Life* movie used for testing. (iii) We combined data from the *The Bourne supremacy* and *The wolf of wall street* movies for training, and tested on the *Life* movie.

**Removal of a single modality features from multi-modal representations.** To remove features for a particular modality $m$ from multi-modal model representations, we rely on a simple method proposed previously by Toneva et al. (2022) and Oota et al. (2023b), in which the linear contribution

of the features to the multi-modal model activations is removed via ridge regression. Specifically, for this ridge regression the feature vector corresponding to modality $m$ is considered as input and the multi-modal representations are the target. We compute the residuals by subtracting the predicted multi-modal feature representations from the actual multi-modal features resulting in the (linear) removal of feature vector for modality $m$ from the pretrained multi-modal embeddings. Because the brain prediction method is also a linear function, this linear removal limits the contribution of features for modality $m$ to the eventual brain alignment. See Fig. 1(B).

**Evaluation Metrics.** We evaluate our models using Pearson Correlation (PC) which is a standard metric for evaluating brain alignment (Jain & Huth, 2018; Schrimpf et al., 2021; Goldstein et al., 2022). Let TR be the number of time repetitions in the test set. Let $Y = \{Y_i\}_{i=1}^{TR}$ and $\hat{Y} = \{\hat{Y}_i\}_{i=1}^{TR}$ denote the actual and predicted value vectors for a single voxel. Thus, $Y \ and \ \hat{Y} \ \in \mathbb{R}^{TR}$. We use Pearson Correlation (PC) which is computed as $corr(Y, \hat{Y})$ where corr is the correlation function.

The final measure of a model's performance is obtained by calculating Pearson's correlation between the model's predictions and neural recordings. This correlation is then divided by the estimated cross-subject prediction accuracy and averaged across voxels, regions, and participants, resulting in a standardized measure of performance referred to as normalized brain alignment. For calculating normalized alignment, we select the voxels whose cross-subject prediction accuracy is $\geq 0.05$.

**Implementation Details for Reproducibility.** All experiments were conducted on a machine with 1 NVIDIA GeForce-GTX GPU with 16GB GPU RAM. We used bootstrap ridge-regression with the following parameters: MSE loss function; L2-decay ($\lambda$) varied from $10^1$ to $10^3$; the best $\lambda$ was chosen by tuning on validation data that comprised a randomly chosen 10% subset from the train set used only for hyper-parameter tuning.

**Statistical Significance.** To determine if normalized predictivity scores significantly higher than chance, we run a permutation test using blocks of 10 contiguous fMRI TRs (considering the slowness of hemodynamic response) rather than individual TRs. By permuting predictions 5000 times, we create an empirical distribution for chance performance, from which we estimate the p-value of the actual performance. To estimate the statistical significance of performance differences, such as between the model's predictions and chance or residual predictions and chance, we utilized the Wilcoxon signed-rank test (Conover, 1999), applying it to the mean normalized predictivity for the participants. In all cases, we denote significant differences (p$\leq$ 0.05) with a $*$ or $\wedge$.

# 6 Results

## 6.1 How effective are multi-modal representations obtained from multi-modal models?

In Fig. 2, we present the average normalized brain alignment scores for both multi-modal and individual modality features. Specifically, we show the normalized brain alignment for cross-modality (ImageBind), jointly pretrained multi-modal (TVLT), and the average from individual video and speech models. The results are shown for whole brain, and also for average across language and visual ROIs. Results for individual ROIs are in Fig. 3.

**Baseline comparison.** To compare the brain predictivity of multi-modal and unimodal models against baseline performance, we employ randomly generated vector embeddings to predict brain activity as baseline. We observe that the brain alignment from a random vector is significantly lower than that of both multi-modal and unimodal models across the whole brain and language-visual processing regions. This shows that the representations from these multi-modal models are significant enough for learning non-trivial alignment with the fMRI recordings of multi-modal stimuli.

**Cross-modal vs. Jointly pretrained multi-modal models vs. Unimodal Models.** Fig. 2(left) displays results for whole brain analysis, where the IB Concat bar plot corresponds to results for representations from a cross-modal model, while TVLT Joint bar plot corresponds to results for representations from a jointly pretrained multi-modal model. From Fig. 2(left), we make the following observations: (i) At the whole brain level, the Wilcoxon signed-rank test shows that the differences in embeddings from the IB Concat and TVLT models are not statistically significant. (ii) The multi-modal embeddings show improved brain alignment compared to unimodal models. Specifically, cross-modal embeddings are significantly better than both unimodal video and speech models, while jointly pretrained embeddings are significantly better than speech models. This implies that cross-

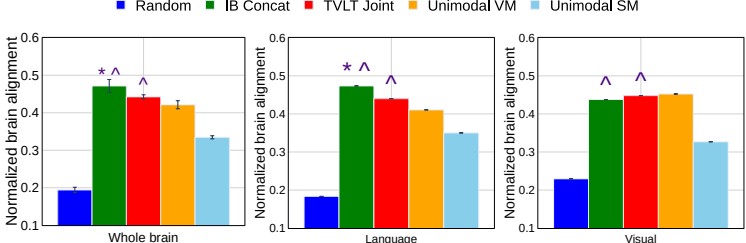

Figure 2: Average normalized brain alignment for both multi-modal and individual modality features across whole brain, language, and visual regions. Error bars indicate the standard error of the mean across participants. ∗ indicates cases where multi-modal embeddings are significantly better than unimodal video models (VM), i.e., p≤ 0.05. ∧, indicates cases where multi-modal embeddings are significantly better than unimodal speech models (SM), i.e., p≤ 0.05.

modal embeddings contain additional information beyond the two modalities, while embeddings from a jointly pretrained model do not provide extra information beyond unimodal visual information but do contain additional information beyond unimodal speech.

We also present average results across language and visual regions in Figs. 2 (middle), and 2(right), respectively. The Wilcoxon signed-rank test shows that the differences in embeddings from the IB Concat and TVLT models are not statistically significant when averaged across language and visual regions. Similar to whole brain performance, in the language regions, cross-modal embeddings are significantly better than both unimodal video and speech models, while jointly pretrained embeddings are significantly better than unimodal speech models. In contrast, for the visual regions, the normalized brain alignment of cross-modal and jointly pretrained embeddings is similar to the performance of unimodal video models. This implies that when we average across visual regions, there is no additional information beyond unimodal video features. However, when compared to unimodal speech features, both multi-modal embeddings show significant improvement.

Since we didn't observe any significant difference at the whole brain level and when averaged across language and visual regions, between cross-modal and jointly pretrained multi-modal models, we attempt to seek if there any any differences when we pay a closer look at the individual ROIs. We present results for language and visual regions such as Angular gyrus (AG), the posterior temporal lobe (PTL), and the inferior frontal gyrus (IFG) in Fig. 3. Additionally, we cover visual regions like early visual cortex (EVC), scene visual areas (SV) and middle temporal gyrus (MT), as well as early auditory cortex (AC). In this figure, we also report the average normalized brain alignment of each modality obtained from multi-modal models. Unlike the whole brain analysis, we observe some differences between cross-modal and jointly pretrained models in several language and visual ROIs. Results for other ROIs are in Fig. 7 in Appendix. Our observations are as follows: (i) Cross-modal IB Concat embeddings are significantly better than TVLT Joint embeddings in semantic regions such as AG and PCC, as well as the multi-modal processing region MT. (ii) Conversely, TVLT Joint embeddings are significantly better than IB Concat embeddings in dmPFC regions. While considering both joint and each modality embeddings from multi-modal models, we make the following observations from Fig. 3: (1) Cross-modal IB video embeddings exhibit improved brain alignment compared to unimodal video in the AG and MT regions with the exceptions of the PTL and AC regions. But this is not the case for IB audio vs unimodal audio. This suggests that video modality information is more relevant and beneficial in the brain for IB Concat embeddings from cross-modality models. (2) TVLT video embeddings show improved brain alignment in the AG, PTL, PCC, dmPFC and EVC regions, with other regions displaying similar normalized brain alignment unimodal video embeddings. (3) Consistent with the cross-modality models, in jointly pretrained TVLT models, TVLT video embeddings significantly outperform TVLT audio embeddings, except in PTL region. These observations indicate that video information is advantageous for both cross-modal and jointly pretrained models, whereas audio embeddings mainly benefit the PTL region.

## 6.2 Which brain regions process uni- and multi-modal information?

From Fig. 3, we observe that multi-modal video embeddings exhibit improved brain alignment not only in the whole brain but also in various language, visual and multi-modal regions. For instance, the cross-modal IB Concat embeddings demonstrate superior brain alignment compared to unimodal video-based models in areas such as the AG, PTL, IFG, and PCC. Moreover, TVLT-joint embeddings

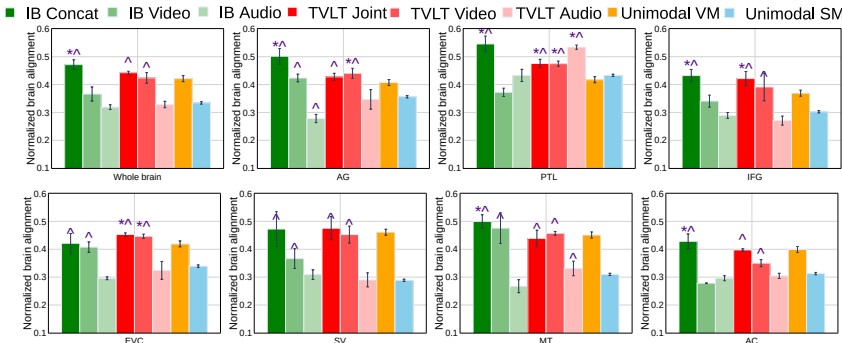

Figure 3: Average normalized brain alignment for video and audio modalities from multi-modal and individual modality features across whole brain and several ROIs of language (AG, PTL and IFG), visual (EVC, SV and MT) and auditory cortex (AC). Error bars indicate the standard error of the mean across participants. ∗ indicates cases where multi-modal embeddings are significantly better than unimodal video models (VM), i.e., p≤ 0.05. ∧ indicates cases where multi-modal embeddings are significantly better than unimodal speech models (SM), i.e., p≤ 0.05.3

show notable enhancements in the AG, PTL, IFG, PCC, dmPFC and EVC regions. In contrast, compared to unimodal speech-based models, all multi-modal embeddings display significantly better brain alignment, except the OV (object visual processing) region. Overall, this observation suggests that integrating multiple modalities leads to transferring information from one modality to another, resulting in improved brain predictability. Based on these, it can be inferred that these multi-modal models can indeed learn multi-modal linkages that are relevant to the brain.

When subjects engage with multi-modality stimuli, we observe that multi-modal embeddings show improvements in semantic regions such as the AG, PCC and dmPFC, and syntactic regions such as the PTL and IFG. Overall, we find that multi-modal information is processed in only a few regions. Furthermore, several regions, including the SV (scene visual area), EVC (early visual cortex), ATL (anterior temporal lobe), IFGOrb, MFG, and dmPFC, exhibit similar brain alignment with both unimodal and multi-modal embeddings.

### 6.3 How is the brain alignment of multi-modal features affected by the elimination of a particular modality?

To understand the contribution of each modality to the multi-modal brain alignment for multi-modal naturalistic stimulus, we perform residual analyses by removing the unimodality features from multi-modal joint representations as well as multi-modal video or audio representations from joint representations and measure the differences in brain alignment before and after removal modality-specific features. Fig. 4 displays the normalized brain alignment for language (AG) and visual regions (MT). We note a decrease in brain alignment for both the AG and MT regions following the removal of video embeddings from cross-modality models, whereas the removal of audio embeddings does not affect the brain alignment. On the other hand, for jointly pretrained models, removal of both video and audio embeddings partially impacts the brain alignment. We observe similar findings for language ROIs such as PTL, MFG, ATL, PCC and visual regions EVC, OV and FV, as shown in Figs. 9 and 10 in Appendix. These results suggest that there is additional information beyond the unimodal embeddings considered in this study that is processed in the visual and language regions.

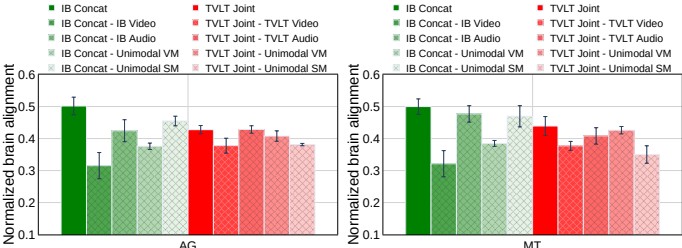

Figure 4: Residual analysis: Average normalized brain alignment was computed across participants before and after removal of video and audio embeddings from both jointly pretrained and cross-modality models. Error bars indicate the standard error of the mean across participants.

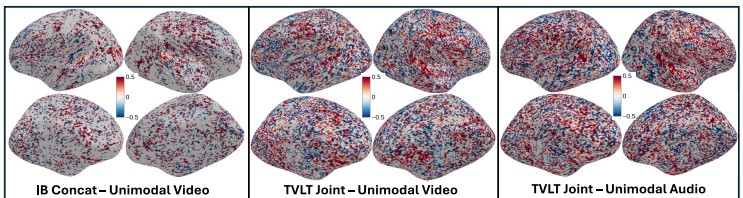

Figure 5: Percentage decrease of brain alignment after removal of (left) Unimodal VM embeddings from IB-Concat (middle) Unimodal VM embeddings from jointly pretrained TVLT, and (right) Unimodal SM embeddings from TVLT Joint. Colorbar indicates the percentage of decrease where red denotes higher and white denotes zero.

**Qualitative analysis.** We compute the percentage decrease in alignment for each voxel following the removal of unimodal video embeddings from the IB Concat (cross-modality) and the TVLT Joint (jointly pretrained model), with projections onto the brain surface averaged across participants, as depicted in Fig. 5. The colorbar shows the percentage decrease in brain alignment, where red voxels indicate a higher percentage decrease and white voxels indicate areas where unimodal video features do not contribute any shared information within the multi-modal context. We observe that removal of unimodal video features leads to a significant drop (40-50%) in performance in the visual regions for IB Concat, and in language regions (PTL & MFG) for TVLT Joint.

# 7 Discussion

Using multi-modal model representations, including both cross-modal and jointly pretrained types, we evaluated how these representations can predict fMRI brain activity when participants are engaged in multi-modal naturalistic stimuli. Further, we compared both multi-modal and unimodal representations and observed their alignment with both unimodal and multi-modal brain regions. This is achieved by removing information related to unimodal stimulus features (audio and video) and observing how this perturbation affects the alignment with fMRI brain recordings acquired while participants are engaged in watching multi-modal naturalistic movies.

Our analysis of multi-modal brain alignment yields several important conclusions: (1) The improved brain alignment of the multi-modal models over unimodal models, across several language, visual, and auditory regions is only partially attributable to the video and audio stimulus features presented to the model. A deeper understanding of these models is required to shed light on the underlying information processing of both unimodal and multi-modal information. (2) Cross-modal representations have significantly improved brain alignment in language regions such as AG, PCC and PTL. This variance can be partially attributed to the removal of video features alone, rather than auditory features. (3) Video embeddings from multi-modal models exhibit higher brain alignment than audio embeddings, except in the PTL and AC regions. This suggests that audio-based models may encode weaker brain-relevant semantics. (4) Both cross-modal and jointly pretrained models demonstrate significantly improved brain alignment with language regions (AG, PCC, PTL and IFG) compared to visual regions when analyzed against unimodal video data. In contrast, when compared to unimodal audio-based models, all multi-modal embeddings display significantly better brain alignment, with the exception of the OV region. This underscores the capability of multi-modal models to capture additional information—either through knowledge transfer or integration between modalities—crucial for multi-modal brain alignment.

**Limitations.** The low alignment scores clearly show that despite the increasing popularity of multi-modal models in tackling complex tasks such as visual question answering, we are still far from developing a model that fully encapsulates the complete information processing steps involved in handling multi-modal naturalistic information in the brain. In the future, by fine-tuning these multi-modal models on specific tasks such as generating captions for videos, we can better leverage their alignment strengths. This approach will allow us to explore task-level brain alignment of three modalities—video, audio, and text—more effectively. Further, multi-modal large language models (MLLMs) (Zhang et al., 2023; Ataallah et al., 2024; Wu et al., 2023) that align visual features from video frames into the LLM embedding space via a trainable linear projection layer, offer promise for enhanced multi-modal capabilities. We would further extend this work by comparing the region-wise brain alignment performance of these multi-modal LLM models with existing approaches.

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

## A  Cross-subject prediction accuracy

We estimate cross-subject prediction accuracy in three settings: (i) training with *The Bourne supremacy* and testing with *Life* data, (ii) training with *The wolf of wall street* and testing with *Life* data, and (iii) training with both *The Bourne supremacy* and *The wolf of wall street* and testing with Life data. We present the average cross-subject prediction accuracy across voxels for the *Movie10 fMRI* dataset and across the three settings in Fig. 6.

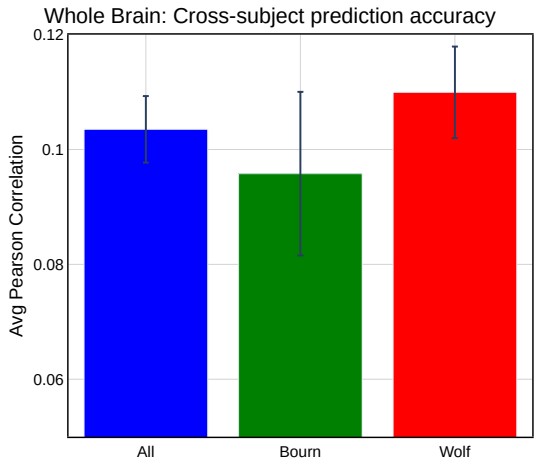

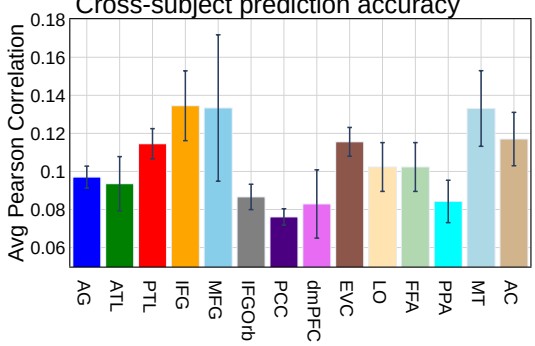

Figure 6: Cross-subject prediction accuracy: (top) across whole brain, (bottom) across language, visual and auditory regions.

## B  Detailed sub-ROIs of language, visual and auditory regions

The data covers seven brain regions of interest (ROIs) in the human brain with the following subdivisions: (i) early visual (EV: V1, V2, V3, V3B, and V4); (ii) object-related areas (LO1 and LO2); (iii) face-related areas (OFA), (iv) scene-related areas (PPA), (v) middle temporal (MT: MT, MST, LO3, FST and V3CD), (vi) late language regions, encompassing broader language regions: angular gyrus (AG: PFm, PGs, PGi, TPOJ2, TPOJ3), lateral temporal cortex (LTC: STSda, STSva, STGa, TE1a, TE2a, TGv, TGd, A5, STSdp, STSvp, PSL, STV, TPOJ1), inferior frontal gyrus (IFG: 44, 45, IFJa, IFSp) and middle frontal gyrus (MFG: 55b) (Baker et al., 2018; Milton et al., 2021; Desai et al., 2022).

## C  Details of pretrained Transformer models

Table 1: Pretrained Transformer-based Encoder Models. All models have 12 layers.

| Model Name | Pretraining |
|---|---|
| Cross-modal Pretrained (ImageBind) | Video & Audio |
| Jointly Pretrained (TVLT) | Video & Audio |
| ViT-B | Image |
| VideoMAE | Video |
| ViViT | Video |
| Wav2Vec2.0-base | Speech |
| AST | Speech |

## D  Effectiveness of multi-modal vs unimodal representations for various brain regions

We now present the results for per unimodal video model and per speech model in Fig. 8. Similar to the average results of unimodal video and speech models, we observe that multi-modal models exhibit better normalized brain alignment than individual unimodal video and speech models across language and visual regions. Among unimodal speech models, the AST model shows better normalized brain alignment than the Wav2vec2.0 model. Among unimodal video models, each unimodal video model displays notably consistent performance across regions.

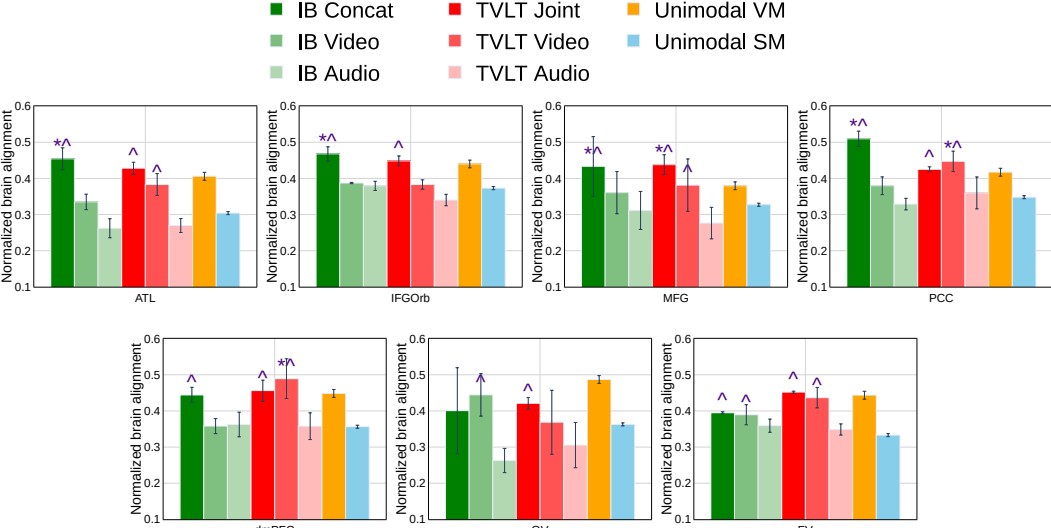

Figure 7: Average normalized brain alignment for per video and audio modalities from multi-modal and individual modality features across whole brain and several ROIs of language (ATL, IFGOrb, MFG, PCC, dmPFC) and visual (OV, FV). Error bars indicate the standard error of the mean across participants.

## E  How is the brain alignment of multi-modal features affected by the elimination of a particular modality?

To understand the contribution of each modality to the multi-modal brain alignment for multi-modal naturalistic stimulus, we perform residual analyses by removing the unimodality features from multi-modal joint representations as well as multi-modal video or audio representations from joint representations and measure the differences in brain alignment before and after removal modality-specific features. Figs. 9 and 10 display the normalized brain alignment for language ROIs such as PTL, MFG, ATL, PCC and visual regions EVC, OV and FV. We note a decrease in brain alignment for these regions following the removal of video embeddings from cross-modality models, whereas the removal of audio embeddings does not affect the brain alignment. On the other hand, for jointly pretrained models, removal of both video and audio embeddings partially impacts the brain alignment.

# F    Layerwise brain alignment

We now plot the layer-wise normalized brain alignment for the Unimodal models and TVLT joint model, as shown in Fig. 11. Observation from Fig. 11 indicates a consistent drop in performance from early to lower layers, specifically for both TVLT joint and unimodal video models. The key finding here is that our results that TVLT joint embeddings showcase improved brain alignment across all the layers compared to unimodal video and speech embeddings.

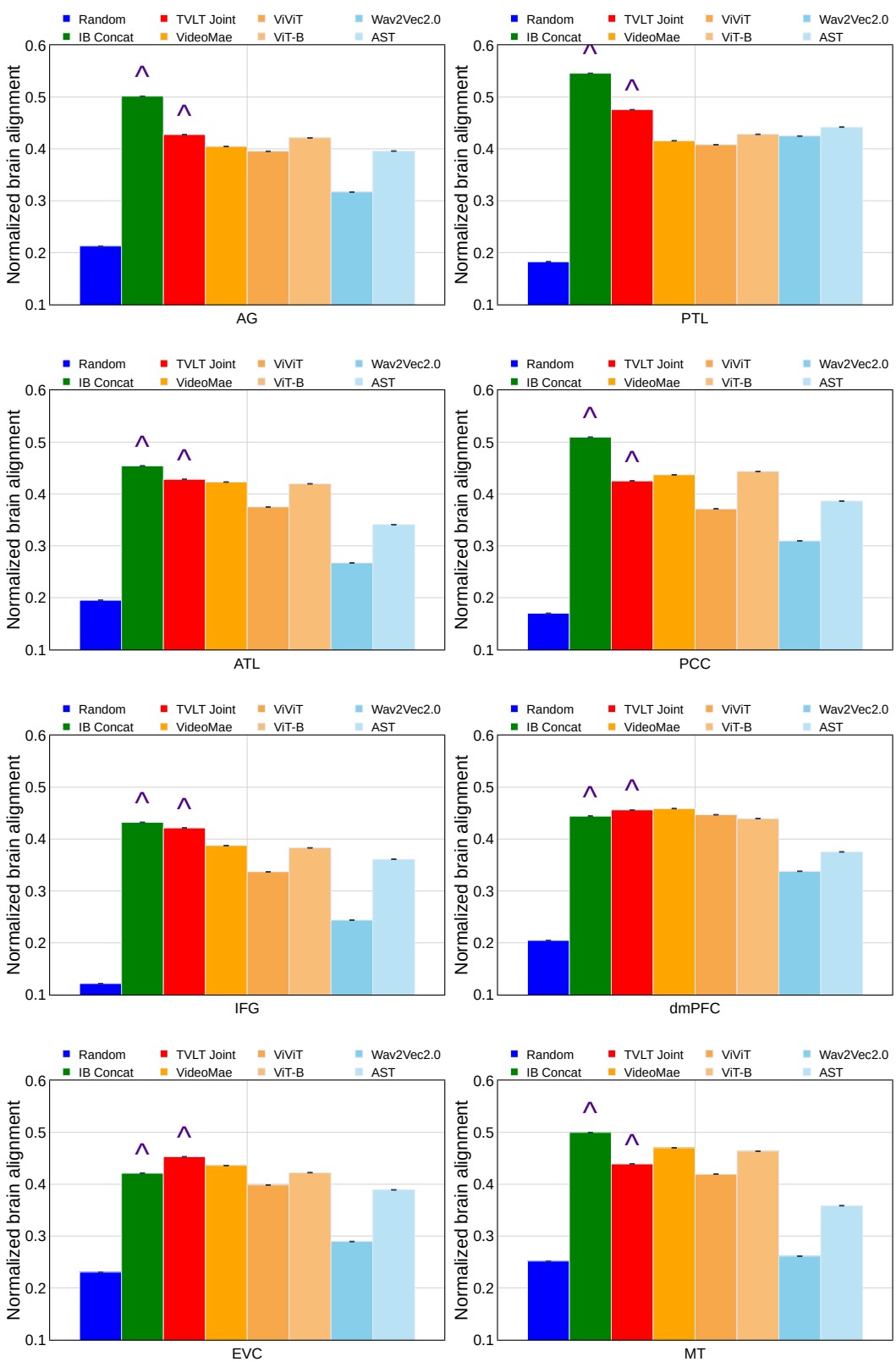

Figure 8: Average normalized brain alignment for video and audio modalities from multi-modal and individual modality features across whole brain and several ROIs of language (ATL, ATL, PTL, IFG, PCC, dmPFC) and visual (EVC, MT). Error bars indicate the standard error of the mean across participants.

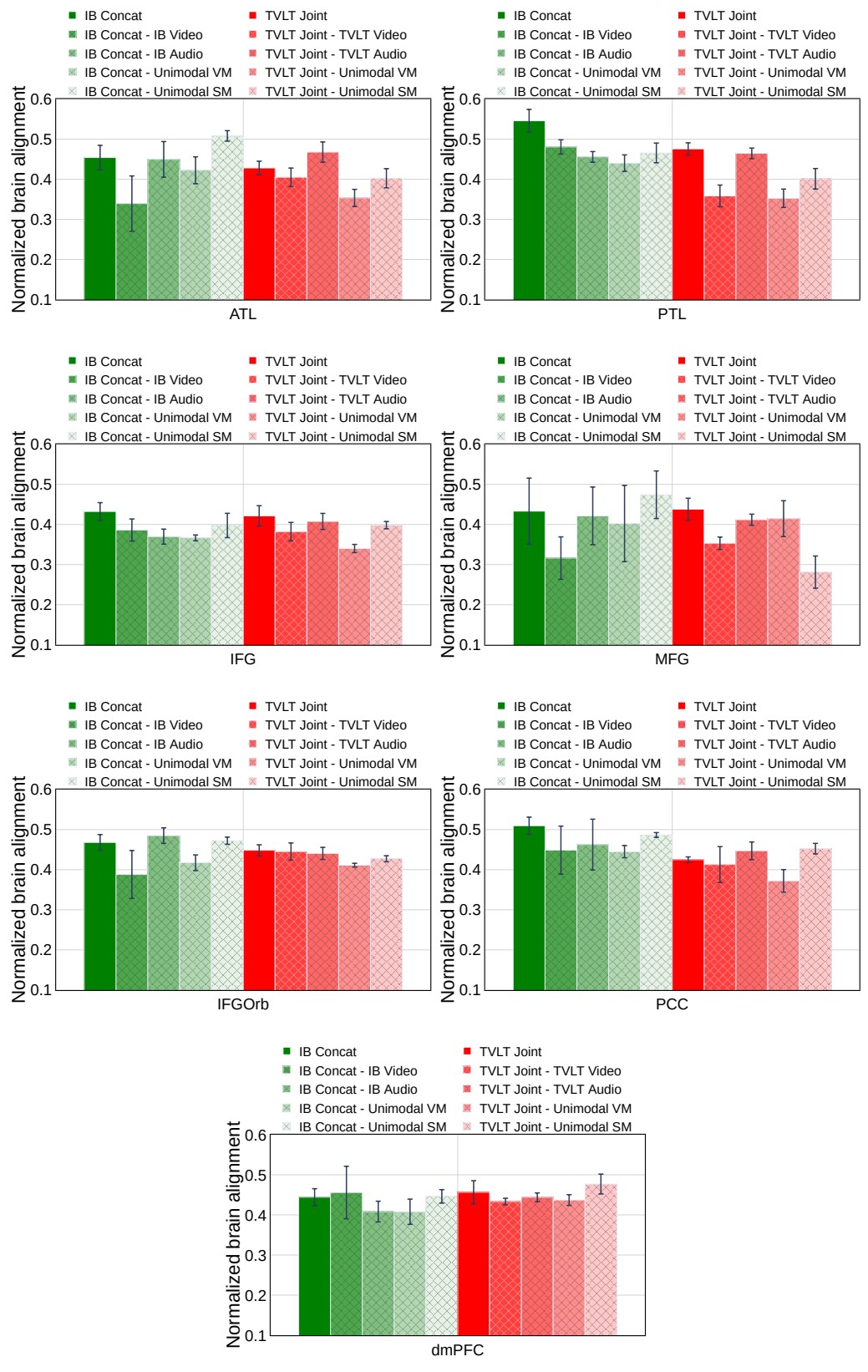

Figure 9: Residual analysis for ATL, PTL, IFG, MFG, IFGOrb, PCC and dmPFC regions: Average normalized brain alignment was computed across participants before and after removal of video and audio embeddings from both jointly pretrained and cross-modality models. Error bars indicate the standard error of the mean across participants.

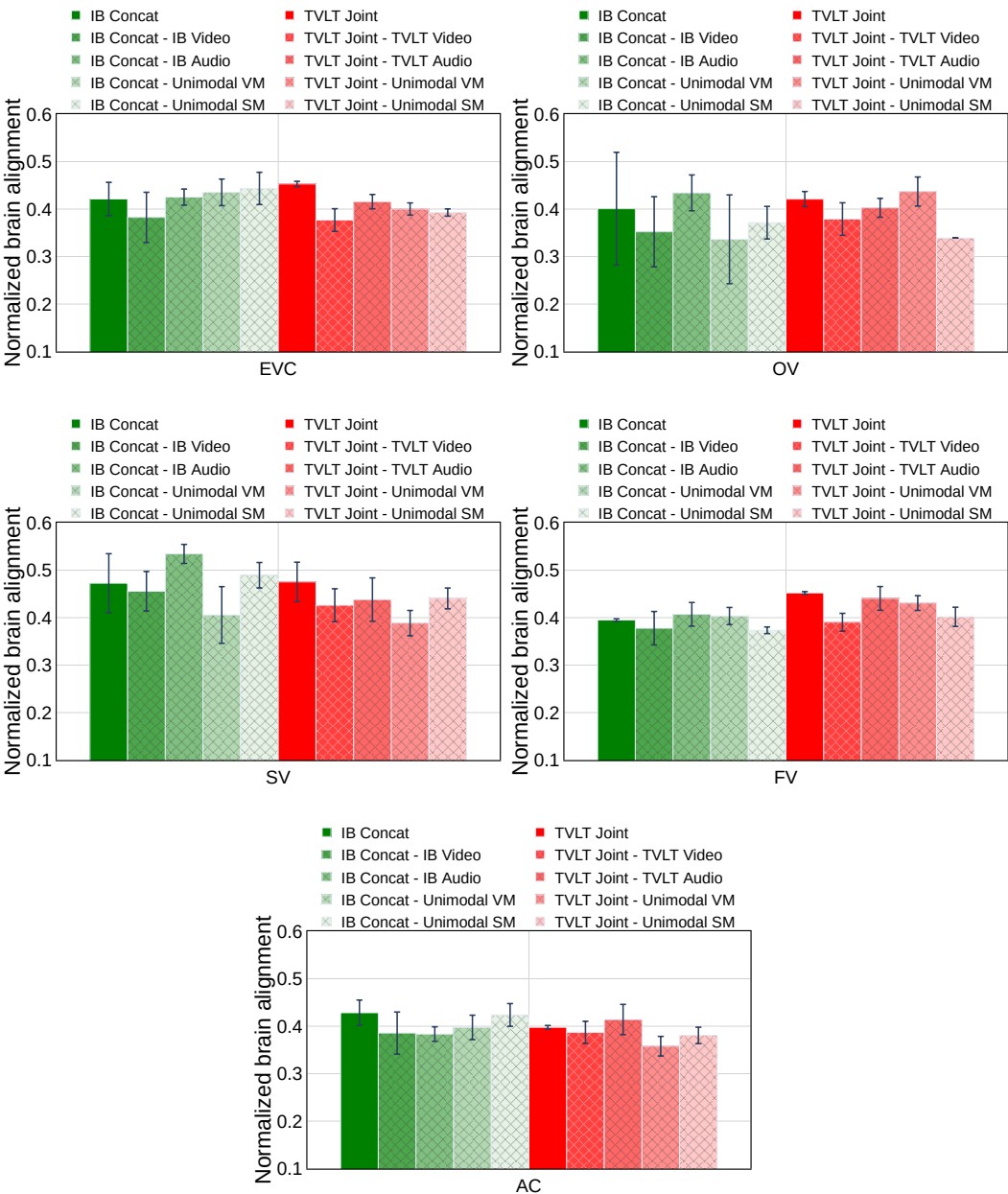

Figure 10: Residual analysis for EVC, OV, SV, FV and AC regions: Average normalized brain alignment was computed across participants before and after removal of video and audio embeddings from both jointly pretrained and cross-modality models. Error bars indicate the standard error of the mean across participants.

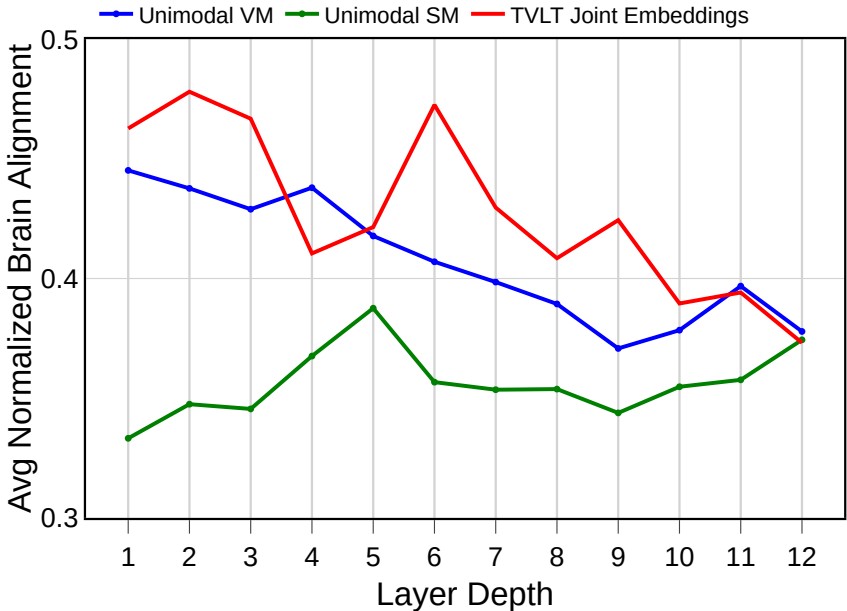

Figure 11: Normalized brain alignment across layers for multi-modal model (TVLT joint embeddings) and unimodal video and speech models.

