# OpenReview forum: "Multi-modal brain encoding models for multi-modal stimuli"
_NeurIPS.cc/2024/Conference — Submitted to NeurIPS 2024_

### Official Review · Reviewer_UHTg · 2024-07-11

**Soundness:** 3
**Presentation:** 3
**Contribution:** 2
**Rating:** 7
**Confidence:** 4

**Summary:**

This paper introduces a novel approach using cross-modal and multi-modal models to align brain activity with naturalistic stimuli, evaluates several unimodal Transformer models, and examines the effects of removing unimodal features from multi-modal representations on brain alignment.

**Strengths:**

- (S1) The paper presents a thorough imaging data analysis, including a detailed description of the dataset, a comprehensive comparison of methods, and a clear summarization of results, which enhances the robustness and transparency of the study.

- (S2) The paper is well-written and clearly presented, making the complex methodologies and findings accessible and easy to understand for readers.

- (S3) The study provides a rigorous comparison of methods, including various method variations, which highlights the strengths and weaknesses of each approach and demonstrates the thoroughness of the analysis.

**Weaknesses:**

- (W1) Figure caption should be more demonstrative and detailed. For example for Figure 1's captions, "Residual analysis" is too vague for the reader to understand the figure. For Figure 5, if the colorbar range from -0.5 to 0.5, then it does not represetn the percentage, but the proportion instead.

**Questions:**

I have two questions regarding to methodology, it will be good if the author can help me clarify it.

(Q1) Given that Pearson Correlation is sensitive to outliers and assumes a linear relationship, how do the authors ensure the robustness of this metric in the context of brain alignment evaluation? Are there any additional statistical measures or preprocessing steps employed to handle potential non-linearities or outliers in the neural data to ensure the reliability of the correlation results?

(Q2) **Permutation Test Configuration**. How do the specific choices in configuring the permutation test, such as using blocks of 10 contiguous fMRI TRs and permuting predictions 5000 times, influence the sensitivity and reliability of detecting significant differences? Are there any potential trade-offs or limitations associated with these choices, particularly considering the hemodynamic response variability across participants?

(Q3) **Wilcoxon Signed-Rank Test Application**. When using the Wilcoxon signed-rank test to compare performance differences, how do the authors account for multiple comparisons or the potential dependency between test conditions?

**Limitations:**

The Limitation section is properly included in the paper.

---

> ### Author Rebuttal · Authors · 2024-08-07
>
> *We thank the reviewer for their strong positive, insightful and valuable comments and suggestions which are crucial for further strengthening our manuscript.*
>
> **1. How do the authors ensure the robustness of Pearson correlation metric in the context of brain alignment evaluation?**
>
> Thank you for this question.
>
> * Our methodology for building encoding models ensures the robustness of the Pearson Correlation metric in the context of brain alignment evaluation by addressing potential outliers and non-linearities through the following steps:
>   - (i) We employ z-score thresholding separately for both Input stimulus representations and brain recordings for training and test datasets. This helps identify and remove extreme outliers that could disproportionately affect the Pearson Correlation results.
>   - (ii) To quantify the model predictions using Pearson correlations, we estimate cross-subject prediction accuracy as studied in previous research. Using this estimated cross-subject prediction accuracy, we measure the normalized brain alignment as Model Predictions / Cross-Subject prediction accuracy *[Schrimpf et al. 2021;Oota et al. 2024;Alkhamissi et al. 2024]*. Using this estimate, we see that these numbers correspond to about 50-70% of the explainable variance by a model representation. Note that we are only averaging across voxels which have a statistically significant brain alignment. We perform the Wilcoxon signed-rank test to test whether the differences between multimodal and unimodal models are  statistically significant. This is explained in lines 249-253 of the paper.
> * As can be seen, Pearson correlation is the most widely used metric to measure brain encoding accuracy. So, we follow the existing literature for this metric choice.
>
> *[1] Schrimpf et al. 2021, The neural architecture of language: Integrative modeling converges on predictive processing. PNAS, 2021*
> *[2] Oota et al. 2024, Speech language models lack important brain relevant semantics, ACL 2024*
> *[3] Alkhamissi et al. 2024,  Brain-like language processing via a shallow untrained multihead attention network, Arxiv 2024.*
>
> **2. Permutation Test Configuration.**
>
> Thank you for this question.
> * Inspired by prior studies [2] [4] [5] [6], we employ the standard implementation of a block permutation test for fMRI data, which involves permuting blocks of 10 contiguous TRs while leaving the order within each block untouched. The choice of these specific configurations is based on established methodologies in previous research.
> * Block Permutation Approach:
>   - Rationale: By shuffling blocks rather than individual TRs, we preserve the temporal structure of the fMRI data (to account for the slowness of the underlying hemodynamic response), which is crucial for maintaining the integrity of the hemodynamic response.
>   - Configuration: We use blocks of 10 TRs, ensuring that the order within each block remains unchanged. This method balances the need to preserve temporal correlations with the need to generate a sufficient number of permutations for robust statistical analysis.
> * Thus, our approach involves shuffling the blocks without averaging over blocks of 10TRs. Specifically, the predictions are permuted 5000 times, and the resulting normalized predictivity scores are used as an empirical distribution of chance performance, from which the p-value of the unpermuted performance is estimated.
> * We will clarify this in the text.
>
> *[2] Oota et al. 2024, Speech language models lack important brain relevant semantics, ACL 2024*
>
> *[4] Deniz et al. 2019, The representation of semantic information across human cerebral cortex during listening versus reading is invariant to stimulus modality. Journal of Neuroscience, 2019*
>
> *[5] Reddy et al. 2021, Can fMRI reveal the representation of syntactic structure in the brain? NeurIPS 2021*
>
> **3. Wilcoxon Signed-Rank Test Application.**
>
> * We performed a Wilcoxon signed-rank hypothesis test for all pairs of models and applied the Benjamini-Hochberg False Discovery Rate (FDR) correction for multiple comparisons.
> * The FDR correction is applied by grouping together all voxel-level p-values across all subjects and choosing one threshold for all the results.
> * This approach helps control the expected proportion of false discoveries among the rejected hypotheses, ensuring the robustness and reliability of our statistical findings. We will update this text and correct this oversight by including it in the final version.
>
> **4. Fig 1 caption should be more demonstrative and detailed.**
>
> Thank you for this suggestion.
> * Detailed caption: we show how residual analysis can be applied to remove unimodal video model features from cross-modal representations for an input X. In the first step, a ridge regression model (r) is trained to map video model (VM) unimodal representations to cross-modal (CM) representations. In step 2, we learn another ridge regression model (g’) to map the residual representation |CM(X)-r(VM(X))| to brain activations.
>
> We will provide more detailed caption in the final version.
>
> **5. Fig5 caption: It is proportion, not percentage.**
>
> * We will correct this typo.

---

> > ### Comment · Reviewer_UHTg · 2024-08-10
> > **Recommendation for Acceptance**
> >
> > Thank you for your response.
> >
> > The authors have addressed my comments and questions clearly. I have no further questions and am inclined to recommend accepting the paper.

---

> > > ### Author Response · Authors · 2024-08-10
> > >
> > > We appreciate the reviewer's feedback and are confident that it has enhanced the paper's quality.

---

### Official Review · Reviewer_4K7X · 2024-07-11

**Soundness:** 2
**Presentation:** 2
**Contribution:** 3
**Rating:** 6
**Confidence:** 4

**Summary:**

The authors present a framework for applying brain encoding models with multimodal stimuli. They apply this to a series of video, audio, and mutlimodal models (cross-modal and jointly embedded models). They introduce a residual analysis to analyze the impact each particular feature had on the corresponding fit in the encoding model. They find that multimodal models significantly out-perform their unimodal counterparts on certain language- and vision-related regions in their fMRI dataset.

**Strengths:**

* Incorporating a new collection of models used for fits to the brain for comparison.
* Expanding to multimodal models, a relatively new space.
* Incorporation of video/speech models, allowing to capture input stimuli over time and removing problems with parsing the stimuli into individual modalities such as ImageBind or VideoMAE.
* Interesting results as seen in Figure 3 showing improvement across several brain regions with multimodal networks. Figure 2 also shows really interesting results with language and visual regions separated.

**Weaknesses:**

* Clarification on feature removal: I think I found the feature removal description in this paper and prior papers a bit confusing and want to ask for some clarification. I wish more space was spent on that in this paper to provide more intuition. I think some extra descriptions would be useful here. See questions.
* In general, I am quite skeptical of how well the feature removal works. For example, there is no guarantee that the features are completely removed in the residual analysis. I would actually like to see a probing analysis to actually establish that the feature is removed.
    * Furthermore, the method of projection is rather confusing. The authors use a regression to “project” unimodal video features (referring to figure 1) into the same space as the multimodal feature space. I don’t think this is necessarily wrong but potentially unreliable without any extra metric to establish how well this works. Having some MSE score or pearson correlation (with the averaged embedding) could help understand how well the projection worked.
    * In my opinion, I wonder why the opposite direction wasn’t taken: instead, project the video features out of the cross-modal/jointly pretrained multimodal representation. You could train a projection matrix to do so using your current vision-language data. To me, this is cleaner and easier to interpret primarily because you aren’t dependent on the quality of your visual representations to capture visual information.
* The paper compares multimodal and unimodal models to demonstrate improvement in brain alignment. One explanation for this improvement could be an improvement in unimodal processing. For example, one interpretation of the current results is that a multimodal model such as TVLT has better visual processing than ViT-B (as an example).  Is this addressed by feature removal? I’m not sure it is. Some extra text to discuss this would be useful. Some extra discussion on model performance would also be useful.
* Baselines
    * The paper doesn’t consider the baseline comparison with randomly initialized models. Why? I think this is a very important baseline for characterizing architectural bias. This was also done in prior works.

**Questions:**

* In lines 85-87, the paper states “alignment… can be partially contributed to the removal of video features alone”. My reading of this is actually as follows: including video features to a speech-only model in a cross-modal fashion improves alignment. Is there a way of rewriting this sentence or maybe adding some context (more of a nit). I think it’s a bit hard for a reader to get an intuition of what feature removal is doing.
* Nit: Could figure 5 be made bigger somehow? This is very hard to read.

**Limitations:**

* I believe these are addressed adequately.

---

> ### Author Rebuttal · Authors · 2024-08-07
>
> *We thank the reviewer for their positive, insightful and valuable comments and suggestions which are crucial for further strengthening our manuscript.*
>
> **1. Clarification on feature removal**
> * For the removal of information from the model representations, we use the previously published approach i.e. direct approach or residual approach.
> * This residual approach directly estimates the impact of a specific modality feature on the alignment between the model and the brain recordings by observing the difference in alignment before and after the specific modality feature is computationally removed from the model representations. This is why we refer to this approach as direct.
> * Additionally, our work is most closely related to that of *Toneva et al. 2022* [1], who employ a similar residual approach to study the supra-word meaning of language by removing the contribution of individual words to brain alignment. Further, this residual approach has been peer reviewed in good venues (Nature Computational Science, NeurIPS, and ACL).
> * To remove features of a particular modality from multimodal model representations, we rely on the direct method as discussed above. In our setting, similar to prior studies, we remove the linear contribution of a unimodal feature by training a ridge regression, in which the unimodal feature vector is considered as input and the multimodal representations are the target.
> * We compute the residuals by subtracting the predicted feature representations from the actual features resulting in the (linear) removal of unimodal feature vectors from pretrained multimodal features. Because the brain prediction method is also a linear function, this linear removal limits the contribution of unimodal features to the eventual brain alignment.
> * Specifically, in Fig. 1B, we show how residual analysis can be applied to remove unimodal video model features from cross-modal representations for an input X.
> * This is done in 2 steps.
>   - In the first step, a ridge regression model (r) is trained to map video model (VM) unimodal representations to cross-modal (CM) representations. In some ways, r(VM(X)) captures that part of the information in CM(X) that can be explained or predicted using VM. Now the job of residual analysis is to check that if we remove this explainable part r(VM(X)) from CM(X) how well can it predict brain activity.
>   - Hence, in step 2, we learn another ridge regression model (g’) to map the residual representation |CM(X)-r(VM(X))| to brain activations. Similarly, residual analysis can also be used to remove unimodal speech features from cross-modal representations for an input X.
>
> *[1] Combining computational controls with natural text reveals aspects of meaning composition, Nature Computational Science 2022*
>
> *[2] Joint processing of linguistic properties in brains and language models, NeurIPS 2023*
>
> *[3] Speech language models lack important brain relevant semantics, ACL 2024*
>
> **2. How well the feature removal works?**
>
> Thank you for raising this question.
> * We provided a more clear description of the feature removal method using residual approach in previous question Q1.
> * We investigate this feature removal method by analyzing how the alignment between brain recordings and multimodal model representations is affected by the elimination of information related to these unimodal features. We refer to this approach as direct, because it estimates the direct effect of a specific feature on brain alignment. Fig 4 (in the main paper) shows that removal of unimodal video features from cross-modal embeddings results in significant drop in brain alignment for the language region AG.
> * To perform probing analysis, unfortunately, there are no available probing tasks for the Movie10 dataset. This is an interesting future direction to annotate tasks for Movie10 dataset and verify the model interpretation via probing before and after removal of unimodal features and also perform several perturbations by doing mechanistic interpretability of models.
>
> **3. Projection method is unreliable without additional metrics like MSE scores or Pearson correlation?**
>
> Thank you for this valuable suggestion.
> * To check the quality of information removal using our residual analysis method, we computed the Pearson correlation scores where unimodal video features are projected onto the multimodal IB Concat feature space using our residual approach.
> * We observe a small Pearson correlation score of 0.555. This low value implies that unimodal video features are successfully removed from multimodal representations.
> * Further, to our knowledge, there is no better alternative to selectively remove information from multimodal models to probe their impact on brain alignment.
>
> **4. Opposite direction of removal: project the video features out of the multimodal representation.**
>
> Thank you for this question.
> * To clarify, in the paper, we have done exactly as you mentioned above. As shown in Fig. 1B (main paper), the residual is calculated as |CM(X)-r(VM(X))| which clearly shows that we have removed video features from the cross-modal/ jointly pretrained multimodal representation.
> * Are you expecting us to perform the experiment in the other direction, i.e., remove multimodal features from unimodal representations? Although that does not sound very reasonable, we are happy to do this if expected.
>
> **5. Do TVLT model has better visual processing than ViT-B, considering feature removal?**
> * Kindly check the rebuttal PDF (Fig 2) & CQ2 response at “Common responses”.
>
> **6. Baseline performance with randomly initialized models.**
> * Based on the reviewers’ suggestion, we now perform experiments with randomly initialized models.
> * Kindly check the rebuttal PDF (Fig 3) & CQ3 response at “Common responses”.
>
> **7. Rewriting of sentence and Fig 5 should be made bigger.**
>
> Thank you for your suggestion. We will update the framing of sentence and important editorial suggestions in the final draft.

---

> > ### Comment · Reviewer_4K7X · 2024-08-10
> >
> > Thank you for the detailed response. I believe you addressed my concerns about clarity. I believe the current experiment in response 3 is sufficient to believe that feature removal is performing reasonably. I would suggest a few small scale experiments as a sanity check.  I'm also pleased to see better improvement from randomly initialized models; In my opinion this is an important point.
> >
> > My only remaining concern is on the interpretation of the comparison of TVLT and ImageBind and the unimodal models as well. In general, I wonder if the results are due to different training/architectural designs as you describe with the other reviewer or some other factor such as dataset or number of parameters. I would suggest augmenting the table in Appendix Section C with more details of the models. I think a discussion of this would raise my score.

---

> > > ### Author Response · Authors · 2024-08-10
> > >
> > > Thank you for your valuable suggestion. We have now augmented the Appendix Table with more details about the models.
> > >
> > > |**Model Name**|**Pretraining Method**|**Number of Parameters**|**Dataset**|**Layers**|**Backbone**|
> > > |-----------|---------|---------|----------------|-----------------|----------|
> > > |ImageBind|  Cross-model multimodal Transformer | ~1.2 B | Audioset, Ego4D, SUN RGB-D | 12 | ViT for Images and Videos, AST for audio |
> > > | TVLT|Jointly pretrained on video & audio (Masked auto encoder)| 88 M | HowTo100M, YTTemporal180M | 12 | ViT for video embeddings, and Spectrogram for audio embeddings|
> > > | ViT-B | Vision Transformer | 86 M | ImageNet | 12 | Transformer encoder |
> > > | VideoMAE| Masked autoencoder for video inputs | 1 B| Kinetics, Epic Kitchens 100, Something-Something v2| 12 | ViT-B
> > > | ViViT| Video vision Transformer| 86 M| Kinetics, Epic Kitchens 100, Something-Something v2 |12 |ViT-B|
> > > | Wav2Vec2.0-base| Speech-based Transformer model | 95 M| Librispeech| 12 | Transformer encoder |
> > > | AST | Audio Spectrogram Transformer| 86 M| AudioSet, ESC-50 & Speech commands | 12 | Initialized with ViT-B weights|
> > >
> > > **Multimodal Models**
> > > * Observation:
> > >    - From the table, we can clearly observe that both multimodal models (ImageBind and TVLT) maintain similar backbone architectures for videos but differ in their backbone architecture for embedding audio as well as in the training strategies. For the TVLT model, video embeddings are captured from the ViT model, while audio embeddings are generated from Mel Spectrograms and jointly pretrained within a single Transformer encoder.
> > >    - In contrast, the ImageBind model uses the ViT model as the backbone for Images and Videos, while the AST model is used for Audio; these individual encoders are used and learn a common embedding space.  Also, the number of parameters and the datasets differ significantly, as you pointed out.
> > >
> > > * Discussion:
> > >   - The model training protocol of TVLT appears more in line with how humans learn during development when they experience multiple modalities simultaneously and the learning is mediated by the experience of joint inter-modal associations. It is unlikely that the human system experiences these modalities in isolation, except in cases of congenital conditions where the inputs from a specific modality are not accessible.
> > >   - Given that the brain alignment observed in TVLT model in a language regions like AG is less sensitive to loss of information from specific modalities, we believe that AG serves as a multi-modal convergent buffer integrating spatio-temporal information from multiple sensory modalities to process narratives *[Humphries & Tibon, 2023]*.
> > >   - The results of high alignment found in AG even in IB-Concat but more brittle with respect to loss of information from a specific modality are also interesting. It would be interesting to study patterns of activation in AG in patients who acquired visual or auditory function later in their life *[Hölig et al., 2023]* to see if one observes such brittleness in the representations acquired.
> > >
> > > *[Humphries & Tibon, 2023] Dual-axes of functional organization across lateral parietal cortex: the angular gyrus forms part of a multi-modal buffering system. Brain Struct Function 228, 341–352 (2023).*
> > >
> > > *[Hölig et al., 2023] Sight restoration in congenitally blind humans does not restore visual brain structure. Cerebral Cortex. 2023;33(5):2152-2161.*
> > >
> > > **Unimodal models**
> > >
> > > * Observation:
> > >   - For the unimodal video models, regardless of their different training strategies and additional pretraining datasets, VideoMAE, ViViT, and ViT-B exhibit similar normalized brain alignment in both language and visual regions.
> > >   - For the unimodal speech models, there are marginal differences in normalized brain alignment between the AST and Wav2Vec2.0 models. However, their performance in the PTL and AC regions is quite similar.
> > >   - This implies that in unimodal models, the differences in brain alignment across language and visual regions are minimal, irrespective of different training strategies or pretraining datasets.
> > >
> > > * Discussion:
> > >   - We have tested with multiple unimodal video models of each type and multiple unimodal audio models of each type, with different objective functions and trained on different amounts of data.
> > >   - We showed that the results we observe generalize within the video- and speech-based model types despite these differences.
> > >   - Still, it is possible that some of the differences in brain alignment we observe are due to confounding differences between model types, and there is value in investigating these questions in the future with models that are controlled for architecture, objective, and training data amounts.
> > >
> > > *We will add this discussion to the final revised manuscript.*

---

> > > > ### Author Response · Authors · 2024-08-12
> > > >
> > > > Dear Reviewer 4K7X,
> > > >
> > > > We appreciate your feedback and effort you have invested in evaluating our work.
> > > >
> > > > In response to your insightful comments, we have addressed the issues you highlighted. We believe these revisions significantly contribute to the clarity and completeness of the paper. We kindly request you to verify our response and consider updating your evaluation score based on the revisions made.
> > > >
> > > > Should you have any further questions or suggestions, we are ready to provide additional information or clarification as needed.
> > > >
> > > > Thanks for your help.

---

> > > > > ### Comment · Reviewer_4K7X · 2024-08-12
> > > > >
> > > > > Thank you for this discussion. I think I'm a bit surprised: I didn't expect such a large difference in the ImageBind parameters and datasets and TVLT. It seems interesting that TVLT does better at all.
> > > > >
> > > > > In general, I believe this paper provides a valuable discussion on how well multimodal models fit activity in the brain, focusing specifically on multimodal stimuli. I also haven't seen people use video models or incorporate audio. However, upon seeing this information, I'm a bit cautious about the conclusions we can draw for example in lines 317-325 and would appreciate more controlled studies. I also would request to add this to the limitations section. I will raise my score to a 6 because I think there is valuable information in this paper.

---

> > > > > > ### Author Response · Authors · 2024-08-13
> > > > > >
> > > > > > Dear reviewer 4K7X,
> > > > > >
> > > > > > Thank you for your positive feedback on our work. We truly appreciate your engagement in the review process and helping us elevate the quality of the manuscript.

---

### Official Review · Reviewer_bruL · 2024-07-12

**Soundness:** 2
**Presentation:** 2
**Contribution:** 2
**Rating:** 4
**Confidence:** 2

**Summary:**

The manuscript investigated the process of multi-modal information in human brains through predicting neural responses based on semantic features extracted by existing models.

**Strengths:**

1. The problem is interesting.
2. The results show insights into brain region's roles in processing multi-modal information.

**Weaknesses:**

## Major
1. The method builds ridge regression based on features extracted by pretrained models. However, I am worried that the findings will be affected by choice of pretrained models. It is important to demonstrate the replication of different pretrained models.
2. For some observations in Section 6, the author only presents the observations and does not give insights based on the observations. For example,
    - What does observation i) in lines 311-312 indicate?
    - What does observation ii) in lines 313-314 indicate?
    - Why is AC an exception for observation (1) in lines 316-317?
    - For observation (2) in lines 320 -322, why is TVLT different from IB-concat, given that both of them contains multi-modal information?
3. Why does the author choose ridge regression instead of more complex machine learning models? Is it possible that more intricate interactions of features extracted by pre-trained models are not captured by a ridge regression model, potentially affecting the results? And if you choose a more complex model, the rank of alignment scores of different models could be altered.
4. I do not know if it is too hard or even impossible, but it would be better to check if the results consistent with some existing neuroscientific findings.
5. In section 6.3, why do IB-concat and TVLT act differently given that they are both multi-modal representations.

## Minor
1. There seems to be a trailing 3 in Fig.3's caption.
2. The author moves the results of some brain regions in Figure 3 to the appendix due to the page limit. Since the author refers to those regions from the main text, it would be better to still include those regions in the main text in my opinion.

**Questions:**

1. The ImageBind model provides 1024 dimensional features, while other models provide 768 features, would it affect the fairness of comparisons between different models?
2. What is $r$ in Figure 1?

**Limitations:**

The authors have adequately addressed most limitations.

---

> ### Author Rebuttal · Authors · 2024-08-07
>
> *We thank the reviewer for their valuable comments and suggestions which are crucial for further strengthening our manuscript.*
>
> **1. Affect of performance by the choice of pre-trained models.**
>
> Thank you for this question.
> * **Impact of Pre-Trained Models**:
>   - To understand the relationship between brain activity and various stimuli, a large body of brain encoding literature over the past decade has utilized a variety of pre-trained language models.
>   - These works have demonstrated that pre-trained models such as BERT, GPT-2, T5, BART, LLaMA, and OPT can predict both text- and speech-evoked brain activity to an impressive degree. Similarly, pre-trained speech models like Wav2vec2.0, HuBERT, AST, and WavLM have shown that speech-based language models better predict auditory cortex activity during speech-evoked brain responses.
>   - Hence, brain encoding studies focus more on interpreting the representations of these models and obtaining insights into brain function rather than the choice of specific pre-trained models.
>   - Prior studies have shown that irrespective of whether the models are encoders, decoders, or encoder-decoder based, they result in similar brain-language alignment.
> * **Our Study:**
>   - In our study, we tested three unimodal vision models and two unimodal speech models and observed similar alignment in brain activity predictions. This consistent performance across different models reinforces the robustness and generalizability of our approach.
>
> **2. What does observation in lines 311-312 & 313-314 indicate?**
>
> *  In Fig 2, we do not observe any significant difference in brain alignment between cross-modal and jointly pretrained models at the whole-brain level or when averaged across language and visual regions.
> * However, in individual language regions, we find that multimodal representations obtained from cross-modal models better predict brain activity in semantic regions such as the Angular Gyrus (AG), Posterior Cingulate Cortex (PCC), and Middle Temporal Gyrus (MTG).
> * This indicates that while both cross-modal and jointly pretrained models perform similarly at a macro level, there are individual differences at micro level. This observation motivated us to do further detailed analysis in Section 6.2 and Section 6.3
>
> **3. Why is AC an exception for observation in lines 316-317?**
>
> * Since AC is an early auditory cortex which processes sound related information unlike higher cognition regions (i.e. language regions), we observe that audio embeddings result in higher degree of brain predictivity than video embeddings. We will include this insight in the final draft.
>
> **4. Why the choice of ridge regression instead of more complex machine learning models?**
>
> Thank you for your question.
> * Since fMRI brain recordings have a low signal-to-noise ratio, and pretrained language models are trained in a non-linear fashion, the model representations are rich and complex.
> * To understand the relationship between brain activity and various stimuli, a large body of brain encoding literature over the past two decades (some papers mentioned below [1-12]) has preferred ridge regression due to its simplicity.
> * Ridge regression is a linear model, making it easier to interpret and understand compared to more complex models. Further, the regularization in ridge regression helps manage the noise effectively, leading to more robust and reliable models.
>
> **5. The results consistent with some existing neuroscientific findings.**
>
> Thank you for raising this point. Many of our findings are in line with previously established neuroscience theories. We describe some of them below.
> * Speech embeddings show better alignment across all language ROIs, but most importantly in the AC region which is related to processing of sound information. This result aligns with previous work which has found that the speech-based language models better predict activations in the early auditory cortex [9] [10] [11].
> * Multimodal embeddings display higher brain alignment in high-level visual processing regions than unimodal models. This observation is consistent with earlier studies which have indicated that the multimodal embeddings from the CLIP model display higher brain alignment than CNN models in high-level visual regions [12].
>
> *[1] Simultaneously uncovering the patterns of brain regions involved in different story reading subprocesses. PLoS One 2014*
>
> *[2] Natural speech reveals the semantic maps that tile human cerebral cortex. Nature 2016*
>
> *[3] Incorporating context into language encoding models for fmri. NIPS 2018*
>
> *[4] Interpreting and improving natural-language processing (in machines) with natural language-processing (in the brain). NeurIPS 2019*
>
> *[5]  The neural architecture of language: Integrative modeling converges on predictive processing. PNAS 2021*
>
> *[6] Brains and algorithms partially converge in natural language processing. Communication Biology 2022*
>
> *[7] scaling laws for encoding models fMRI. NeurIPS 2023*
>
> *[8]  Self-supervised models of audio effectively explain human cortical responses to speech. ICML 2022*
>
> *[9] Toward a realistic model of speech processing in the brain with self-supervised learning. NeurIPS 2022*
>
> *[10] Joint processing of linguistic properties in brains and language models. NeurIPS 2023*
>
> *[11] Speech language models lack important brain relevant semantics, ACL 2024*
>
> *[12] Incorporating natural language into vision models improves prediction and understanding of higher visual cortex, Nature Machine Intelligence 2023.*
>
> **6. Why do IB-concat and TVLT act differently?.**
>
> Kindly check response CQ4 at “Common responses”.
>
> **7. Questions**
> * r refers to the ridge regression model.
> * Given that the underlying encoder models are the same and only the projection results in 1024 dimensions for the ImageBind, the difference in feature dimensionality should not significantly affect the fairness of comparisons.
>
> **8. Minor**
> * We will address these minors.

---

> > ### Comment · Reviewer_bruL · 2024-08-09
> > **Official Comment by Reviewer bruL**
> >
> > Thanks for the rebuttal. I still have major questions regarding your CQ4 to the question 6.
> > I understand that TVLT and IB-concat are distinct regarding model architecture and training strategy. However, since the behaviors of the two models are different in lines 350-353, could you provide some insights about the root cause of it?

---

> ### Author Response · Authors · 2024-08-10
>
> Thank you for your question regarding CQ4 and the distinct behaviors observed between TVLT and IB-concat in lines 350-353. We appreciate your attention to this important aspect of our work.
>
> **IB-Concat**
>
> * For cross-modality models, the alignment in regions AG and MT is extremely high, and this alignment is only partially explained by video features. This implies that significant unexplained alignment remains after the removal of video features. Conversely, the removal of speech features does not lead to a drop in brain alignment, indicating that there is additional information beyond speech features that is processed in these regions.
> * This means that in cross-modality models, when transferring knowledge from one modality to another, the model relies more heavily on visual information. As a result, the model becomes more focused on video inputs rather than audio inputs. This likely reflects the model’s preference for using the detailed visual features that align closely with brain activity in regions AG and MT, leading to the observed high alignment.
>
> **TVLT**
> * For jointly pretrained multimodal models, the alignment in regions AG and MT is extremely high, and this alignment is partially explained by both video and audio features. Unlike cross-modal representations, the TVLT model learns a more balanced representation of both video and audio features. This leads to integrated information from both modalities, making the model less sensitive to the loss of features from a specific modality.
> * As a result, we observe only a small drop in brain alignment when either modality is removed. This suggests that the model is capturing more high-level abstract and semantic information that goes beyond the specific features of just one modality.
>
> **Additional discussion:**
>   - The model training protocol of TVLT appears more in line with how humans learn during development when they experience multiple modalities simultaneously and the learning is mediated by the experience of joint inter-modal associations.
>   - It is unlikely that the human system experiences these modalities in isolation, except in cases of congenital conditions where the inputs from a specific modality are not accessible.
>   - Given that the brain alignment observed in TVLT model in a language regions like AG is less sensitive to loss of information from specific modalities, we believe that AG serves as a multi-modal convergent buffer integrating spatio-temporal information from multiple sensory modalities to process narratives *[Humphries & Tibon, 2023]*.
>   - The results of high alignment found in AG even in IB-Concat but more brittle with respect to loss of information from a specific modality are also interesting. It would be interesting to study patterns of activation in AG in patients who acquired visual or auditory function later in their life *[Hölig et al., 2023]* to see if one observes such brittleness in the representations acquired.
>
> *[Humphries & Tibon, 2023] Dual-axes of functional organization across lateral parietal cortex: the angular gyrus forms part of a multi-modal buffering system. Brain Structure Function 228, 341–352 (2023).*
>
> *[Hölig et al., 2023] Sight restoration in congenitally blind humans does not restore visual brain structure. Cerebral Cortex. 2023;33(5):2152-2161.*
>
> Should you have any further questions or suggestions, we are ready to provide additional information or clarification as needed. We kindly request you to verify our response and consider updating your evaluation based on the revisions made.
>
> Thanks for your help

---

> > ### Author Response · Authors · 2024-08-12
> >
> > Dear Reviewer bruL,
> >
> > We appreciate your feedback and effort you have invested in evaluating our work.
> >
> > In response to your insightful comments, we have addressed the issues you highlighted. We believe these revisions significantly contribute to the clarity and completeness of the paper. Additionally, other reviewers have recognized the comprehensive comparison of methods and the clear summarization of results, which we feel strengthens the robustness and transparency of our study.
> >
> > We kindly request you to verify our response and consider updating your evaluation score based on the revisions made.
> >
> > Should you have any further questions or suggestions, we are ready to provide additional information or clarification as needed.
> >
> > Thanks for your help

---

> > > ### Comment · Reviewer_bruL · 2024-08-12
> > > **Official Comment by Reviewer**
> > >
> > > I appreciate the author's response and adjust my score accordingly.

---

> > > > ### Author Response · Authors · 2024-08-13
> > > >
> > > > Dear reviewer bruL,
> > > >
> > > > Thank you for your consideration and for adjusting your score. We appreciate your feedback and are grateful for the opportunity to address your concerns.
> > > >
> > > > We would like to highlight some novel contributions from our computational experiments:
> > > >
> > > > * Our computational modeling experiments on brain alignment point out that there are two routes to establishing multimodal representations in the brain – one through simultaneous (joint) training (TVLT) and the other through modality-specific training, followed by cross-modal learning of joint representation (IB-Concat).
> > > > * While both models yield similar, high brain alignment results, their sensitivities to loss of unimodal information are distinctly different. While TVLT-based brain alignment is less sensitive to loss of unimodal information, IB-Concat-brain alignment degrades with loss of unimodal inputs.
> > > > * These interesting differences can now be verified by Cognitive Neuroscientists, taking these as novel testable predictions for designing future empirical experiments. Thus, the computational experiments lead to novel testable predictions.
> > > >
> > > > We hope these contributions will be considered in your evaluation.
> > > >
> > > > Regards,
> > > >
> > > > Authors

---

> > > > > ### Author Response · Authors · 2024-08-14
> > > > >
> > > > > Dear Reviewer bruL,
> > > > >
> > > > > As the author-reviewer discussion phase is set to close in 11 hours, we kindly request that you review our response and consider updating your evaluation score, as you previously mentioned.
> > > > >
> > > > > We greatly appreciate your time and consideration.
> > > > >
> > > > > Best regards,
> > > > > The Authors

---

> > > > > > ### Comment · Reviewer_bruL · 2024-08-14
> > > > > > **Official Comment by Reviewer brUL**
> > > > > >
> > > > > > I am confused by the message because I have already increased the score from 3 to 4. Is there anything I missed?

---

> > > > > > > ### Author Response · Authors · 2024-08-14
> > > > > > >
> > > > > > > Dear Reviewer bruL,
> > > > > > >
> > > > > > > Thank you for your prompt reply. It seems there was a misunderstanding on our part regarding your evaluation. If you are still considering our response, we appreciate your careful review. Otherwise, we are grateful for your feedback and the updated evaluation score.
> > > > > > >
> > > > > > > Best regards,
> > > > > > > The Authors

---

### Official Review · Reviewer_28FZ · 2024-07-12

**Soundness:** 2
**Presentation:** 2
**Contribution:** 2
**Rating:** 5
**Confidence:** 5

**Summary:**

This paper addresses an important question of how accurately multi-modal models can predict brain activity when participants are engaged in multi-modal stimuli. The key challenge is how to integrate or separate the information from different sensory modalities. This work explored two types of models, ie cross-modal and joint pretrained models. Through extensive experiments, this paper found some things that are important to unveil the brain encoding principles, which are important to the AI community.

**Strengths:**

The paper writing good, and research problems are well explained.
The encoding pripline is clearly illustrated.
Experimental designs are insightful.

**Weaknesses:**

- My major concern is about the train-test settings. There exist `clock’ (temporal) relationship which might lead to information leakage during inference. This paper did not mention how to advoid such an issue.

- The data collection process should be blocked to aviod inter-data correlation, espeically for joint-modal training. The three settings mentioned in the paper do not really account for the speciality of brain signals.

**Questions:**

- For the cross-modal setting using ImageBind, which only takes a single modality during inference stage, the alignement has already done by ImageBind.  Therefore, this setting does not actually make a difference from uni-modality setting as ImageBind can not achive the modality combintion. Therefore, an anysis is needed regarding the performance with and without alignment of video and audio or language.

- Contraditory to the claim in abstract and introudction, language is actually not a sensory modality. In this case, lanuage is similar to audio which can be measured by sensor. For this reason, the conclusion “Both cross-modal and jointly pretrained models demonstrate significantly improved brain alignment with language regions” is somehow questionable. More detailed analysis is needed to further clarify this claim.

**Limitations:**

See above comments

---

> ### Author Rebuttal · Authors · 2024-08-07
>
> We thank the reviewer for their strong positive, insightful and valuable comments and suggestions which are crucial for further strengthening our manuscript.
>
> **1. There exist `clock’ relationships in train-test settings lead to information leakage during inference.**
> * We made sure to follow proper practice for our training and testing settings, taking care that data leakage doesn’t happen.
> * Please note that for the main results reported in the paper (as mentioned on lines 157-160)  data from two movies “The Bourne supremacy” and “The wolf of wall street” movies was used for training, while testing was done on a third movie: “Life”.
> * Thus, the train and test sets are totally disjoint and the model can’t use any clock relationships from the training data during inference. To be completely clear: independent encoding models are trained for each subject using data concatenated from two movies (The Bourne supremacy: 4024 TRs and The wolf of wall street: 6898 TRs). The test set consisted only data from the "Life" movie (2028 TRs). Thus there is no possibility of any information leakage during inference on the test set.
> * The training data followed contiguous TRs, in line with prior studies where multiple stories are combined in a contiguous fashion for training, and a separate story is used for testing. This method of combining training data follows established protocols in the field. Since an entirely different movie is used for testing, our results are robust and free from temporal information leakage.
>
> **2. The data collection process should be blocked to avoid inter-data correlation. The three settings do not really account for the speciality of brain signals.**
> * Blocking/ Inter-data correlation: If we understand this issue correctly, we have addressed this in our answer to the previous question. If not, perhaps the reviewer can explain what he means by this question.
> * To be clear, the training and testing sets are completely disjoint in the voxelwise encoding model. The data collection process is explained in detail in Section 3. This process is blocked to avoid all kinds of inter-data correlation for both unimodal as well as joint-model training. The original Courtois NeuroMod dataset creators have already ensured that data from different subjects and modalities are collected independently.
> * In all three settings, testing is always on the third movie, “The Life”. This allows us to evaluate (1) the generalization of models across different training datasets, and (2) test the impact of increasing the number of TRs in the training dataset on brain prediction results. Therefore, the three settings account for generalization of models rather speciality of brain signals.
>
> **3. Cross-modal setting using ImageBind: with and without alignment of video and audio**
>
> Thanks for asking this interesting question.
> * During inference, ImageBind does not require all modalities to be present simultaneously. We can provide data from just one modality (e.g., audio) and the model will still function correctly by finding related embeddings in other modalities (e.g., images, text) within the same embedding space.
> * This flexibility allows for a variety of applications, such as cross-modal retrieval and classification, without needing to input all modalities at once.
> * However, it is not true that this setting does not differ from the uni-modality setting, because as mentioned previously, even if we pass a single modality to ImageBind, it retrieves the relevant embeddings from other modalities from its aligned embedding space, so we don’t need to explicitly align the video and audio.
> * If we provide two modalities to the ImageBind, then the retrieval from ImageBind would be conditioned on the explicit signal for the other modality that we are providing, so we are limiting the exploration scope of ImageBind to search for aligned embeddings of other modalities.
> * To investigate the cross-modal setting with and without alignment of video and audio, we created a plot (Fig 1 in the rebuttal PDF) comparing the normalized brain alignment for three regions AG (angular gyrus), SV (scene visual) and MT (Middle Temporal). This plot shows that with alignment improves brain predictivity for the video modality across all three regions, while the audio modality shows improved alignment in the MT region, with the other regions maintaining similar performance.
> * Kindly also check the rebuttal PDF at “Common responses”.
>
> **4. The conclusion “Both cross-modal and jointly pretrained models demonstrate significantly improved brain alignment with language regions” is somehow questionable. More detailed analysis is needed.**
>
> Thank you for raising this concern. We agree with the reviewer and in fact we never claimed that language is a sensory modality.
> * Our intuition in the abstract and introduction pertains to the hierarchical nature of processing of information via early sensory regions (early visual and early auditory), and then on to higher cognitive processing regions, including language areas.
> * While the extant literature focused on brain alignment with unimodal data (including studies with incongruent unimodal inputs), the current study investigates brain alignment when information from multiple modalities is utilized – trained either in a cross-modal fashion or in a joint-modality setting.
> * Further in order to compare with the existing results, we undertook experiments with additional unimodal settings.
> * Consequently, all our brain region results include visual, auditory, as well as language regions.
> * Our results seem to point out that multiple-modal models seem to capture additional variance than unimodal models (see Fig. 3 in main paper) both in the sensory regions as well as in the language regions.
> * In this context, we feel that our conclusion, "Both cross-modal and jointly pretrained models demonstrate significantly improved brain alignment with language regions," is reasonable and appropriate.

---

> ### Comment · Reviewer_28FZ · 2024-08-12
>
> The rebuttal addressed most of my concerns. I keep my original rating.

---

> ### Author Response · Authors · 2024-08-12
>
> Dear Reviewer 28FZ,
>
> We appreciate your feedback and are confident that it has enhanced the paper's quality.
>
> Should you have any further questions or suggestions, we are ready to provide additional information or clarification as needed. We kindly request you to consider updating your evaluation (score) based on the revisions made.
>
> Regards,
>
> Authors

---

> > ### Author Response · Authors · 2024-08-14
> >
> > Dear Reviewer 28FZ,
> >
> > As the author-reviewer discussion phase is set to close in 11 hours, we want to express our gratitude for your engagement. If you are satisfied with our response, we kindly request that you consider updating your evaluation score based on the revisions made.
> >
> > We greatly appreciate your time and consideration.
> >
> > Best regards,
> > The Authors

---

### Author Rebuttal · Authors · 2024-08-07

*We thank the reviewers for their strong positive, insightful and valuable comments and suggestions which are crucial for further strengthening our manuscript.*

**CQ1. Cross-modal setting using ImageBind: with and without alignment of video and audio? (reviewer 28FZ)**

* To investigate the cross-modal setting with and without alignment of video and audio, we created a plot (Fig 1 in the rebuttal PDF) comparing the normalized brain alignment for three regions AG (angular gyrus), SV (scene visual) and MT (Middle Temporal).
* This plot shows that with alignment improves brain predictivity for the video modality across all three regions, while the audio modality shows improved alignment in the MT region, with the other regions maintaining similar performance.

**CQ2. Do TVLT model has better visual processing than ViT-B, considering feature removal? (reviewer 4K7X)**

Thank you for this question.
* The comparison of normalized brain alignment for video and audio modalities from multi-modal and individual modality models across whole brain and several language and visual rois in Appendix Fig 8.
* Based on the reviewers' suggestion, we now compare the visual processing of the TVLT model and ViT-B using the feature removal method (Fig 2 in rebuttal pdf).
* We didn’t observe any significant difference in brain alignment at the whole brain level and when averaged across visual regions between TVLT and ViT-B model.
* However, in the individual language and visual regions, we observe that TVLT models display significantly improved brain alignment in language regions (PCC, IFGOrb, MFG, IFG, PTL, ATL and AG) (Fig 8 in Appendix) and visual regions including EVC, SV, FV (Fig 2 in rebuttal pdf).
* From these plots, our findings indicate that even after removing ViT-B features from the TVLT model, the TVLT model still shows improved brain alignment compared to unimodal models.
* This suggests that the improvement is not solely due to better unimodal processing but also due to the effective integration of information from multiple modalities.
* This analysis suggests that the observed improvements in brain alignment are due to both the enhanced unimodal processing capabilities and the effective integration of multimodal information.
* We will include additional text in the final manuscript to discuss these findings and provide a more comprehensive analysis of model performance.

**CQ3. Baseline performance with randomly initialized models. (reviewer 4K7X)**

* Based on the reviewers’ suggestion, we now perform experiments with randomly initialized models for ImageBind, TVLT, Unimodal VM and Unimodal SM (Fig 3 in rebuttal pdf).
* Using these results, we find that randomly initialized models show significantly better alignment than random vectors.
* However, the pretrained model embedding brain alignment is significantly better than randomly initialized models. Fig 3 in rebuttal pdf shows whole brain alignment results with random vectors, randomly initialized models and their corresponding pretrained models.
* Clearly, pretrained models > randomly initialized models > random vectors.

**CQ4. Why do IB-concat and TVLT act differently given that they are both multi-modal representations. (reviewer bruL)**

Thank you for this question.

* We would like to clarify that although both IB-concat and TVLT are designed to handle multi-modal information, their differing approaches to architecture, training methodology, and information handling result in distinct behaviors.
* In IB-concat, there are two separate encoders for each modality. These encoders are trained independently. After training, the representations from these two encoders are contrasted and mapped into a shared space. This process involves the transfer of knowledge from one model to the other.
* On the other hand, jointly pretrained multimodal models like TVLT integrate information from different modalities earlier in the processing pipeline, allowing for more intricate interactions between modalities throughout the model. This can lead to richer, more integrated multi-modal representations.
* Hence, the representations from these two multi-modal models are different due to their distinct approaches to training and integration.

---

### Decision · Program_Chairs · 2024-09-25

**Decision:**

Reject

**Comment:**

1. Reviewers asked questions about the interpretability of the results given the vast differences in dataset, model size, and model architecture. How much of the difference is accounted for by the multimodal vs unimodal nature of the task vs these other differences? The number of parameters varied by several orders of magnitude, the sizes of the datasets moreso. It's hard to interpret the results in light of this.

2. Reviewers also raised a concern about the statistical tests employed. Was the Wilcoxon signed rank test multiple comparisons corrected? Authors provided an answer that yes it was and that they would update the manuscript to reflect this. Some information is still lacking, such as how the per-region results were comparison corrected.  The manuscript reports results on varying numbers of regions in the main manuscript and in the appendix. Some of which are referenced as key results in the main manuscript and which reviewers wanted included in the main body. What was actually included in the multiple comparisons correction for the per-region results if one was performed is unclear. This was only a minor point.

3. By far the two most important issues brought up by reviewers were: the very notion of what is a modality and what these results mean. These two issues are intertwined. Confusion in the first resulted in confusion in the second.

The notion of what a modality is and what is being investigated is confusing throughout the manuscript starting with figure 1 and the language on page 2. The manuscript divides multimodal models into two categories:

> Multi-modal models are of two broad types: (i) cross-modal pretrained models, where first individual modality encoders are trained and then cross-modal alignment is performed, and (ii) jointly pretrained models, which involve combining data from multiple modalities and training a single joint encoder.

A model that takes as input only vision is multimodal by this standard. From the point of view of ML we call the resulting embeddings multimodal because they're useful for multimodal tasks. But from the point of view of neuroscience this makes little sense. How can a model that is a function of only vision be considered multimodal?

Alternatively, would the authors call a region of the brain whose entire activity can be explained purely by visual input and nothing else "multimodal"? This is doubtful.

In that case, referring to IB as a multimodal model doesn't make sense in this context. The regions where it dominates are not multimodal, they are unimodal: their activity is a function of only one input.

It would be far simpler to read the manuscript if the authors changed the language they used to: unimodal, unimodal aligned, and multimodal. Where unimodal/multimodal refers to whether the output of a model is a nonlinear function of a single modality or multiple modalities. Graphs could be labeled in this way instead of by model.

This would also make the results much easier to interpret, the other major issue reviewers had. Every question in the manuscript has a complex series of answers that are difficult to follow. Each focuses on different graphs, with different models, different regions, etc. Confusingly many graphs list whether models are better than the unimodal speech models even though that's not the question being asked. For example, "6.1 How effective are multi-modal representations obtained from multi-modal models?" has a 23 line answer with references to two different figures, one of which is only in the appendix, with the answer being broken down into 3 other questions that compare different sets of models and report only a small subset of regions where those models are effective. It's extremely hard to understand what to do with this answer.

It would be much simpler to understand the results if the manuscript provided 1 table. For every region the alignment of every model.

Then, every question could be formulated as a simple function of this table. Every question could be posed in language and then the precise translation into what that means in terms of the table could be provided. Each question would have 1 simple answer, a list of regions. Every question could also present a map of the brain with relevant areas highlighted.

The precise questions the manuscript asks would then be simple to understand and these could be tied back to figure 1 reusing the same notation.

The relationship between questions is also very hard to understand without such a systematic way of reporting results. In particular question 6.3 (how does feature removal affect the alignment?) ends in a qualitative analysis. It could instead show a table of rank order differences for models across regions. If the feature removal matters, this is where it would be seen.

4. Separately, reviewers also raised concerns about the feature elimination experiment. Authors contend that since other publications perform such experiments they are well founded. But their own results show that feature elimination doesn't work as expected. Authors remove ViT-B features from TVLT joint embeddings and improve brain alignment! If feature elimination worked as intuition expected --- that it removed something from the feature space of another model, then this should be impossible (modulo cases where data is too limited). Linear regression should ignore unrelated features, unless the original model had too many degrees of freedom and it overfit, removing features should not lead to better alignments.

In any case, feature removal is unnecessary for any of the author's or reviewer's questions. For example, one reviewer asks "The paper compares multimodal and unimodal models to demonstrate improvement in brain alignment. One explanation for this improvement could be an improvement in unimodal processing. For example, one interpretation of the current results is that a multimodal model such as TVLT has better visual processing than ViT-B (as an example). Is this addressed by feature removal? I’m not sure it is. Some extra text to discuss this would be useful. Some extra discussion on model performance would also be useful. "

The authors then report the feature removal experiment. Instead, they could simply use TVLT as they already do and only provide it with only visual input masking audio and compare against ViT-B. The authors already even have these results in the manuscript, but presented in such a piecemeal way that I cannot back out the answer to this question. This is another advantage of reformatting into a more systematic way of reporting results, point 3 above: one can ask new questions of the same data post-hoc.

Overall: The authors have done the essential experiments for this publication, and have even organized them in figure 1, but then the rest of the manuscript doesn't have a clear structured way of presenting results and experiments. Only piecemeal results are shown and the precise meaning of questions isn't tied back to figure 1. I encourage the authors to simplify their presentation, to make it more thorough by presenting their data plainly so that readers can make up their own minds (both in a table and rendered on top of a brain). This could be a good submission, it just needs some reorganization. As it stands, it's impossible to understand the answers to the interesting questions that the manuscript engages with.